# Genomic regions under selection in the feralization of the dingoes

Shao-jie Zhang [1,2,5], Guo-Dong Wang[1,3,5], Pengcheng Ma[1,5], Liang-liang Zhang[4], Ting-Ting Yin[1], Yan-hu Liu[1], Newton O. Otecko[1], Meng Wang[2], Ya-ping Ma[2], Lu Wang[2], Bingyu Mao[1,3*], Peter Savolainen[4*] & Ya-ping Zhang [1,2*]

Dingoes are wild canids living in Australia, originating from domestic dogs. They have lived isolated from both the wild and the domestic ancestor, making them a unique model for studying feralization. Here, we sequence the genomes of 10 dingoes and 2 New Guinea Singing Dogs. Phylogenetic and demographic analyses show that dingoes originate from dogs in southern East Asia, which migrated via Island Southeast Asia to reach Australia around 8300 years ago, and subsequently diverged into a genetically distinct population. Selection analysis identifies 50 positively selected genes enriched in digestion and metabolism, indicating a diet change during feralization of dingoes. Thirteen of these genes have shifted allele frequencies compared to dogs but not compared to wolves. Functional assays show that an A-to-G mutation in ARHGEF7 decreases the endogenous expression, suggesting behavioral adaptations related to the transitions in environment. Our results indicate that the feralization of the dingo induced positive selection on genomic regions correlated to neurodevelopment, metabolism and reproduction, in adaptation to a wild environment.

[1] State Key Laboratory of Genetic Resources and Evolution, Kunming Institute of Zoology, Chinese Academy of Sciences, Kunming 650223, China. [2] State Key Laboratory for Conservation and Utilization of Bio-resource in Yunnan, Yunnan University, Kunming 650091, China. [3] Center for Excellence in Animal Evolution and Genetics, Chinese Academy of Sciences, Kunming 650223, China. [4] KTH Royal Institute of Technology, School of Engineering Sciences in Chemistry, Biotechnology and Health, Department of Gene Technology, Science for Life Laboratory, SE-171 65 Solna, Sweden. [5] These authors contributed equally: Shao-jie Zhang, Guo-Dong Wang, Pengcheng Ma *email: mao@mail.kiz.ac.cn; savo@biotech.kth.se; zhangyp@mail.kiz.ac.cn

Domestication is the process when a wild species is bred in captivity and modified by artificial selection, becoming phenotypically and genetically distinct from the wild ancestor[1–7]. Feralization is, in a sense, the reverse process, when a domestic species escapes human control, adapts to the wild through natural selection, and diverges from the domestic ancestor into a genetically distinct population[8]. In the shift from artificial to natural selection, feralization is accompanied by phenotypic changes resulting in a phenotype closer to that of the original wild ancestor than to the domestic type. For instance, feralized rodents tend to look more like wild than domestic rodents[9], feral chicken on the island Kauai have increased brooding like wild Red Junglefowl[10], and the dingo's hunting and social behavior is more similar to that of the wolf than of the dog[11–13]. In plants, weedy rice (a feralized rice population) has a closer semblance to wild than to domestic rice for several growth characters[14–16].

Although the feralization process has aroused considerable research interest, only limited research about the genomic mechanisms involved in this phenomenon has so far been presented. A major obstacle for such studies is that, in most cases, the feral populations are not isolated from the wild and/or domestic ancestors, implying a problem to distinguish genetic change caused by feralization from change caused by crossbreeding with the ancestral populations. So far, only two comprehensive studies of genomic changes under feralization have been performed, on feral chicken on Kauai, and on Chinese weedy rice[10,15]. The research on feral chicken shows adaptation of genes associated with sexual selection and reproduction but suggested that feralization and domestication mostly target different genomic regions[10]. Similar conclusions are reached concerning the feralization of Chinese weedy rice[15], suggesting convergent evolution of different weedy types but little overlap of genes under selection in the domestication and feralization processes. However, both these studies have problems to distinguish genetic change caused by feralization from change caused by crossbreeding with the ancestral populations.

The dingo (*Canis dingo*) is a wild canid native to Australia, and its apex predator[17]. It originates from domestic dogs but has, since it arrived at least 3500 years ago, developed into a phenotypically and genetically distinct population of feral dogs. Its appearance is similar to the domesticated dog but there are big differences in its behavior[18,19]. Like the wolf, the dingo is a predominantly meat-eating omnivorous animal, and lacks the expansions of the alpha-amylase locus giving improved starch digestion in dog lineages that are associated with agrarian societies[12,20,21]. Although not a variant associated with domestication[22], it differentiates dingoes and wolves from most domestic dogs. Dingoes hunt in the wild, can catch and kill large prey such as kangaroos, cattle, water buffalos, and wild horses, and use the same tactics as their wild ancestor, wolves, to hunt the large prey[11]. While young dingoes are often solitary, adults often form a settled group, and the dingo's social behavior is as flexible as that of a coyote or gray wolf[13,23]. The dingo is an ideal and unique model for studying the evolutionary and genomic mechanisms of feralization, because of two features. Firstly, the dingo population has a longer history of feralization than any other animal, since their arrival in Australia at least 3500 years ago[24,25]. Secondly, the dingoes have been isolated from both their domestic and their wild ancestor during this feralization except the last 200 years, because of Australia's position outside the natural range of wolves and its isolation until the arrival of Europeans. Today, dingoes and European breeds hybridize, especially in the Southeast, but in most other regions hybridization is limited[26–29]. Therefore, unlike the feral chickens of Kauai island and weedy rice, the dingoes have not experienced hybridization with ancestral populations which may complicate the deciphering of the genomic mechanisms of feralization.

In the present study, we sequence the nuclear genomes of 10 dingoes from across Australia and 2 New Guinea Singing Dogs (NGSDs; wild canids living in highland New Guinea), and retrieve the genomes of a worldwide representation of 78 dogs and 21 wolves from literature. Based on this, we analyze population structure and phylogenetic structure to assess the detailed demographic history and migration route of dingoes. The results show dingoes originated from domestic dogs that migrated to Australia approximately 8300 years ago. We perform selection scans to decipher the genomic mechanisms of natural selection under feralization and to reveal the correlation between selection in domestication and feralization.

## Results

**Sample collection and whole-genome sequencing**. Ten dingoes and two NGSDs were sequenced for the current study. The samples of dingoes have a wide distribution across Australia (Fig. 1a), and the two NGSDs are from the NGSD Conservation Society stud book. After DNA extraction, individual genomes were sequenced to an average of 14.7×. We also retrieved 97 canine whole-genome sequences from published articles[3,21,30–33], which involved 1 dingo, 1 Taiwan village dog, 43 indigenous dogs from China and Vietnam, 19 individuals from various breeds, 4 village dogs from Africa, 6 Indian village dogs, 3 village dogs from Indonesia, 3 village dogs from Papua New Guinea, and 21 wolves from across Eurasia (Supplementary Data 1). Downloaded data have a high quality and an average sequencing depth of 14.6×. Overall, the dataset covers all major dog and wolf groups[34] that are putative ancestors of dingoes. Raw sequence reads were mapped to the dog reference genome (Canfam3) using the Burrows-Wheeler Aligner (BWA)[35]. DNA sequence analysis was done using the Genome Analysis Toolkit[36]. After strict filtering, we identified ~24.7 million autosomal SNPs for further analysis (see details in the Methods).

**Population structure and phylogenetic analysis**. Principal component analysis (PCA) of the 109 individuals was performed to explore the relationships among dingoes, NGSDs, and other canids. In a two-dimensional plot of the genotypes, there is a clear separation in three groups: wolves, dogs and dingoes/NGSDs. Dogs can be divided into two basic groups: dogs from Europe and indigenous dogs from Asia. All dingoes and NGSDs cluster together tightly, on a relatively large distance from the dogs. Thus, the dingo and NGSD populations are genetically clearly distinct from domestic dogs. Among the dogs, Indonesian village dogs cluster closest to the dingoes/NGSDs, followed by indigenous dogs from southern East Asia (South China) (Fig. 1b). We then analysed the 10 dingoes and 2 NGSDs separately, to explore their detailed structure (Fig. 1c). The two-dimensional plot separates NGSDs from the dingoes. The dingoes cluster together, but are distributed in three sub-clusters in accordance with geographical origin: Southeast, West/central and Northeast Australia. This suggests that there are subpopulations within the dingo population.

To explore the genetic relationships among the 109 individuals, we also performed a structure analysis using the expectation maximization (EM) algorithm in ADMIXTURE to cluster the individuals into different numbers of groupings[37]. Partitioning the individuals into four groups gave least cross-validation error (0.31132), and separated the samples into: (i) wolves, (ii) dingoes and NGSDs, (iii) indigenous dogs from southern East Asia and Indonesia, and (iv) breeds and village dogs from other regions

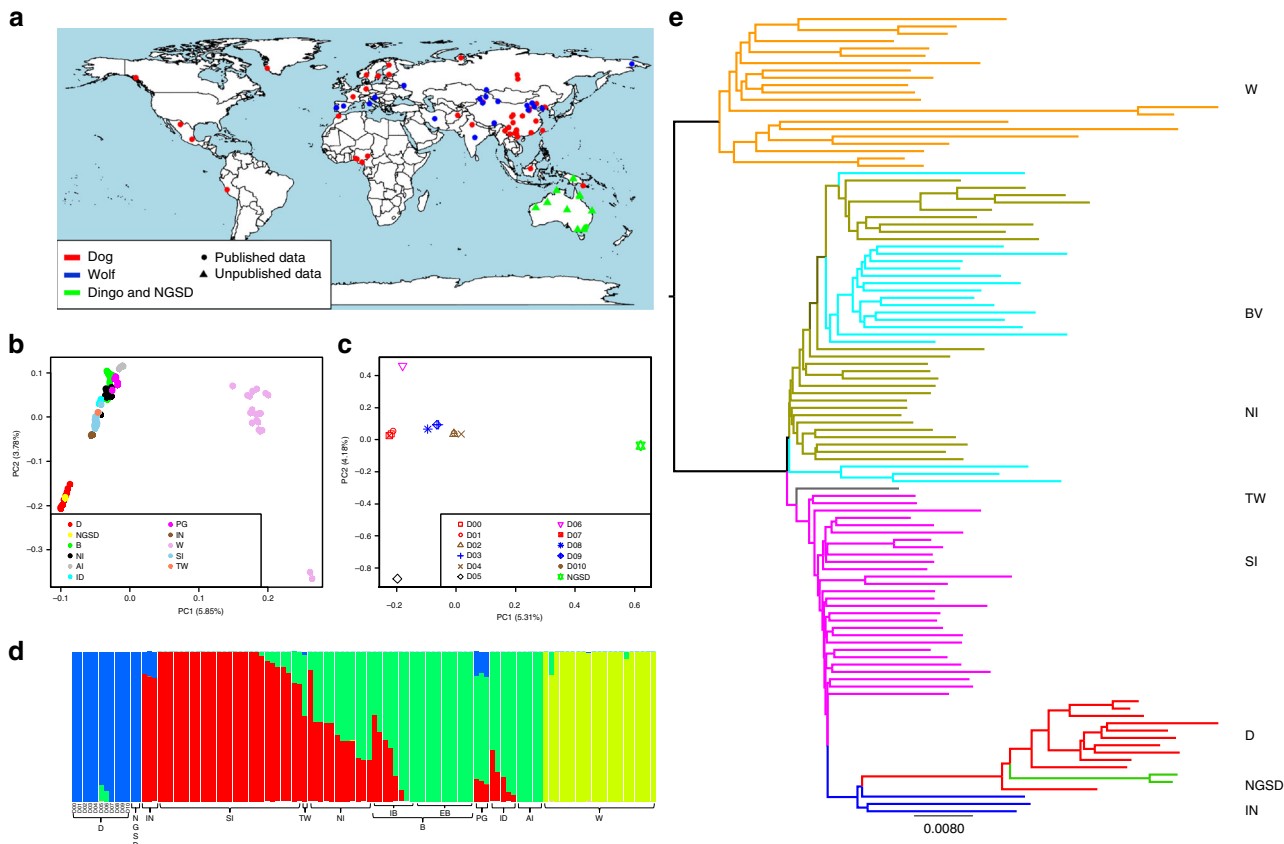

**Fig. 1 Population structure and genetic diversity of 109 canids. a** Geographic locations of the 109 canids analyzed in this study. The map was drawn by the R Packages (maps: https://CRAN.R-project.org/package = maps). **b** Principle component analysis of the 109 canids. **c** Principle component analysis of only the dingo and NGSD, red = dingo from Southeast Australia; blue = dingo from West/central Australia; brown = dingo from Northeast Australia; green = NGSD. **d** Structure analysis of all the 109 individuals. **e** A phylogenetic tree for all the 109 individuals. W, wolves (orange in tree); BV, breeds and village dogs outside China/SE Asia (sky blue in tree); NI, indigenous dogs from north China (green brown in tree); TW, Taiwan indigenous dog (gray in tree); SI, indigenous dogs from southern China (purple in tree); D, dingoes (red in tree); NGSD, New Guinea Singing Dogs (green in tree); IN, Indonesian village dogs (deep blue in tree);B, breeds; IB, intermediate breeds; EB, European breeds; PG, Papua New Guinea village dogs; ID, Indian village dogs; AI, African village dogs.

(Fig. 1d, Supplementary Figs. 1, 2). We find an admixture of these four components with varying proportions among indigenous dogs from northern China, some dog breeds and the village dogs from India and New Guinea, consistent with the results of the PCA. Notably, two dingoes, D05 and D06, show a mixture indicating hybridization with European breed dogs, and D05 originate from the region in Australia with highest incident of dingo-dog hybridization (the Southeast)[26–29]. Therefore, we performed D-statistics analysis using qpDstat in the Admixtools software package[38] to test events of gene flow between the dingoes and European breeds in the form of D (Dhole, European breed; Pop1, Pop2), where Pop1 was all dog groups tested in turn and Pop2 was each individual dingo. Interestingly, the results varied depending on the tested Pop 1 (Supplementary Fig. 3). When Pop1 was NGSDs, there were seven dingoes showing significantly positive D (Z>3): (D00 (Z = 8.457), D01 (Z = 9.394), D03 (Z = 5.338), D05 (Z = 14.551), D06 (Z = 10.221), D07 (Z = 8.888), and D08 (Z = 5.423)). However, when Pop1 was any of the other populations (Indonesian village dogs, indigenous dogs from southern China, Taiwan indigenous dog, indigenous dogs from north China, Indian village dogs, African village dogs, respectively), there was significantly negative D (Z < −3) in all cases (Supplementary Fig. 3). We then used NGSDs as Pop2, testing D (Dhole, European breed; Pop1, NGSDs), where Pop1 was all dog groups and the dingoes tested in turn. This gave

significantly negative D in all cases (Indonesian village dogs: −23.2, indigenous dogs from southern China: −27.8, Taiwan indigenous dog: −34.1, indigenous dogs from north China: −42.0, Indian village dogs: −34.7, African village dogs: −28.0 and dingoes: −8.8), but it is notable that dingoes had considerably less negative value than all other populations. This indicates that all dog populations show higher affinity to European breed dogs than the dingoes, except the NGSDs which have the lowest affinity to European breeds. Therefore, we made another D-statistics analysis, where Pop1 was all dingo individuals tested in turn and Pop2 was all other dingo individuals tested in turn. This showed that when three of the dingoes (D01, D05, and D06) were Pop2, they had significantly positive D (Z > 3) compared to most other dingoes and no significantly negative D (Z < −3) in any comparison (Supplementary Data 2), suggesting that these three dingoes could have a gene flow with European breeds. We also performed an additional D-statistics using the red fox[33] as outgroup and obtained very similar results (Supplementary Data 2, Supplementary Fig. 4).

We further performed phylogenetic analysis by the Neighbor-Joining (NJ) approach (Fig. 1e, Supplementary Fig. 5). The result matches the observations from the PCA and structure analysis. First, dogs and dingoes/NGSDs separate from the wolves, and then they further split into two clades, one including dingoes and NGSDs together with indigenous dogs from Indonesia, southern

East Asia and Taiwan, and the other including village dogs and breeds from all other regions. Indonesian village dogs are closest to the dingoes and NGSDs, and dogs from southern East Asia are basal to the clade. This suggests that indigenous dogs from southern East Asia may be the ancestors of dingoes. Notably, India has been suggested as a possible origin for the dingo but, similarly to the PCA, the dogs from India cluster in the second clade, far from the dingoes[39,40]. We also performed phylogenetic analysis by the Maximum-Likelihood approach (Supplementary Fig. 6), obtaining consistent results. We further used the qp3pop program[38] to perform outgroup f3-statistics analysis in the form of f3(Dingoes, Pop2; Dhole)[38,41], to assess the relative genetic similarity of the dingo population to the other populations (Supplementary Table 1). The highest value of the f3-statistics (indicating highest degree of shared genetic history) was obtained for the NGSD population, followed by Indonesian dogs, and indigenous dogs from southern China. We also performed TreeMix analysis (Supplementary Figs 7, 8). The topology is consistent with the aforesaid phylogeny constructed by the NJ and ML approaches, and indicates a single admixture event: from the dingo/NGSD clade to the Papua New Guinea village dog lineage. All these results agree in suggesting that indigenous dogs from southern East Asia were the ancestors of dingoes.

Notably, the branch to dingoes and NGSDs is relatively long. We therefore estimated nuclear diversity using the genetic diversity $\theta_\pi$, grouping individuals into six dog populations. The result shows that dingoes had the lowest diversity of the six dog populations (Supplementary Fig. 9). This suggests a severe bottleneck event in the evolutionary history of dingoes, or long periods of isolation with low effective population size, which may explain the long phylogenetic branch[39,42].

**Demographic and migration histories.** Based on the results from the TreeMix, phylogeny, and Outgroup f3 analyses, the indigenous dogs from southern East Asia are plausible ancestors of dingoes and NGSDs, with the Indonesian village dogs as the most closely related population. To study the migration and demographic history of the dingoes, we performed a demographic analysis using G-PhoCS[43]. We repeated the computations three times by randomly picking 1000 neutral loci and randomly selecting three samples among the dingoes (Supplementary Table 2) and set the gene flow between southern East Asia dogs and Indonesian dogs. Based on a mutation rate of $1.3 \times 10^{-9}$ per site per year[44] and a generation time of 3 years[40–42], our analysis indicates that the split between dingoes and Indonesian village dogs occurred around 8300 (CI: 5400–11,200) years ago and that, before that, Indonesian village dogs diverged from the indigenous dogs from southern East Asia around 9900 (CI: 6500–12,700) years ago (Fig. 2a, Supplementary Data 3). Furthermore, we

performed a second round of G-PhoCS analysis replacing the two dingoes indicated to be admixed with dogs (D05 and D06) with D01 and D08, respectively (Supplementary Table 3), giving results consistent with the first analysis (Supplementary Data 4). The G-PhoCS analysis also shows that the dingo population has a very small effective population size compared to the dog populations. We also used smc++ employing unphased whole genomes to infer population history[45]. This analysis approximated the split between Indonesian village dogs and dingoes at around 9100 years ago, in consistence with the result of G-PhoCS. We also used smc++ to estimate dates for the population history of dingoes/NGSDs and dogs. The result shows that the dog population experienced a slight growth after the population split, while the dingo and NGSD populations suffered a decrease (Fig. 2b), followed by an increase possibly reflecting the expansion into the new ecological niches in Australia and New Guinea. Notably, the NGSDs show a severe decrease in more recent times followed by a sharp increase. This is consistent with the history of the western population of NGSDs (bred outside New Guinea the last 60 years), which originates from very few individuals.

**Mitochondrial genome analysis.** We also performed phylogenetic analysis based on mitochondrial genomes, analyzing totally 35 dingoes and 3 NGSDs, the 10 dingoes and 2 NGSDs sequenced in this study and 25 dingoes and 1 NGSD from Cairns et al.[46], in the context of 169 dogs and 8 wolves from across the Old World from Pang et al.[47]. We constructed a phylogenetic tree showing all dingoes and NGSDs to group into a single branch, separated from all domestic dogs except one, a dog (A103 10002) originating from Hunan in South China (Fig. 3a). The dingo/NGSD branch is part of the major domestic dog haplogroup A, to which approximately 75% of domestic dogs worldwide belong. Haplogroup A has six sub-haplogroups, and the dingo/NGSD branch is part of sub-haplogroup a2, which is frequent in dogs originating from across East Asia but absent in western Eurasia[47]. Notably, of the eight dogs clustering closest to the dingo/NGSD branch, seven were from Mainland or Island Southeast Asia and one from East Siberia. These results indicate that dingoes and NGSDs originate from domestic dogs in Southeast Asia, via Island Southeast Asia, and that dingoes and NGSDs are closely related, as earlier suggested[24,42,46,48].

To study the detailed phylogeny among dingoes and NGSDs we created a sub-dataset including all individuals in the dingo/NGSD branch and the three most closely related dogs (yellow box in Fig. 3a), and constructed new phylogenetic trees (Fig. 3c, Supplementary Figs. 10, 11). These trees show a division of dingoes into two main branches, following a geographical distribution earlier reported by Cairns et al.[46]; all dingoes from Southeast and East Australia (we denote this region S/E), except

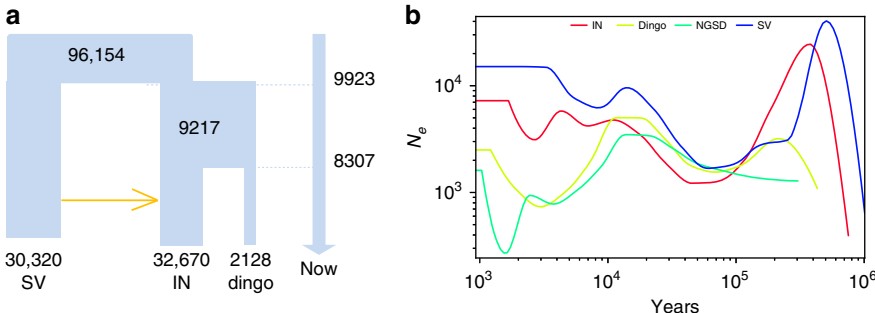

**Fig. 2 Demographic history of dingoes and dogs from southern East Asia. a** Demographic history inferred for indigenous dogs from southern China (SV), Indonesian village dog (IN) and dingo using G-phocs. **b** Inferred effective population sizes over time for indigenous dogs from southern China (SV), Indonesian village dog (IN), dingo and NGSD using SMC++.

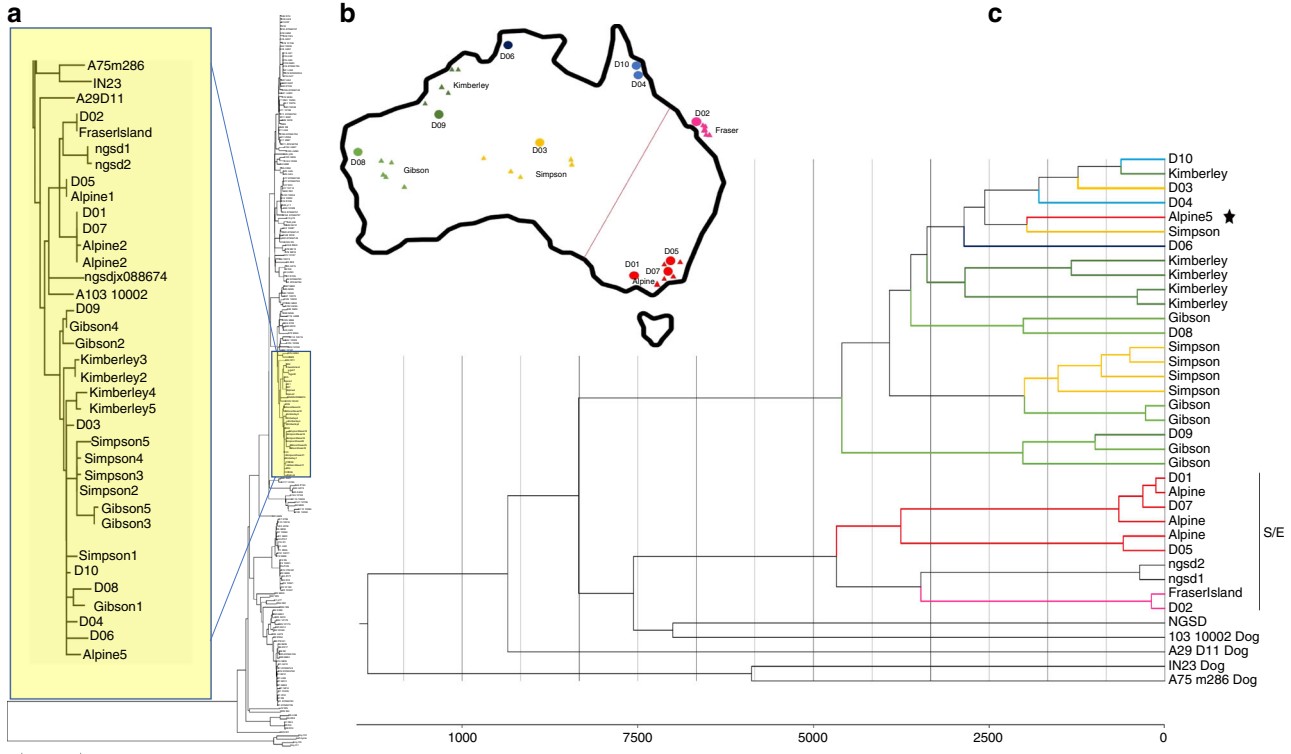

**Fig. 3 Phylogenetic and demographic history analysis of mtDNA. a** Neighbor-joining tree based on mitochondrial genomes from 35 dingoes and 3 NGSDs, and from 169 domestic dogs and 8 wolves from across the Old World, with 4 coyotes as outgroup. The yellow box and inset figure indicate the branch in which all dingoes and NGSDs cluster together with a single domestic dog from South China, and the 3 most closely related dogs outside this branch. **b** Map depicting geographic sampling of dingoes across Australia. Circles represent the 10 individuals sequenced in this study and triangles 25 additional samples from Cairns et al. The red line indicates the genetic subdivision between the southeastern/eastern part (S/E) and the rest of the continent. **c** Bayesian analysis of mitochondrial genomes for the sub-dataset identified in Fig. 3. The dingo/NGSD branch including all dingoes and NGSDs and a single South Chinese domestic dog and, as outgroup, the three most closely related dogs outside that branch. The scale axis indicates time estimates using the mutation rate of $7.7 \times 10^{-8}$ per site per year with SD $5.48 \times 10^{-9}$ from Thalmann et al.[49]. The colored branches indicate geographical origin of dingo samples, see Fig. 2. The star highlights the single dingo sample from southeast Australia that does not cluster in the S/E-related branch.

one, group in one branch, and all dingoes from all other parts of Australia group in the other (Fig. 3b, c). The S/E-related branch also includes two of three NGSD samples, while the third NGSD and the domestic dog from South China (A103 10002) have an intermediate position. Notably, outside the S/E region there is only limited geographical structure among the dingoes. Thus, there is a genetic subdivision of dingoes between the southeastern/eastern part of Australia and the rest of the continent.

Molecular clock analysis (based on a mutation rate of $7.7 \times 10^{-8}$ per site per year)[49] suggests a most recent common ancestor (MRCA) for all dingoes and NGSDs (the division into the two main branches) ~8300 years ago, in agreement with the nuclear genome estimate for the split between dingoes and Indonesian dogs. Notably, the two main branches both have MRCAs ~4600 years ago, indicating population expansions.

**Natural selection in feralization**. Our analyses of population structure and demography confirms that the dingoes originated by feralization of domestic dogs around 8300 years ago and have remained virtually isolated from both the wild and the domestic ancestor until recent time. This affirms that the dingo is an excellent model for studies of the genomic effects of feralization. We used analysis of population branch statistics (PBS)[50] and iHS[51] to identify positive selection in the dingoes. Firstly, the PBS was calculated by the formula (Eq. 1, see Methods for details), with non-overlapping 20 kb genomic windows. By comparing the

three pairwise Fst val PBS1 $= \frac{T_{ds} + T_{db} - T_{bs}}{2}$ ues, we can estimate the frequency change that occurred in the dingoes[50]. We retrieved genomic regions with the top 5% PBS1 windows by the value of PBS1 >0.14476. Furthermore, we performed a windowed iHS test[52], dividing the genome into the same non-overlapping 20 kb windows, and identified candidate regions for selection as those in which more than 30% of the sites had an iHS absolute value above the threshold (2.4217, top1% of iHS). In summary, the overlap of the two approaches indicated 87 candidate windows under positive selection, containing 50 genes (Supplementary Data 5) considered as candidates associated with feralization of dingoes.

We performed the GO enrichment evaluation using the parent-child model[53] in the topGO R package[54]. To control for biasing factors, such as gene size and clustering of related gene families, we used the same approach as Pendleton et al.[22]. We calculated permutation-based $p$ values ($p_{perm}$) for each GO term, and the parent-child significance scores observed for each GO term were compared with the distribution of identified gene sets by applying the parent-child test by 1000 randomly permuted genome intervals. Hereby, we identified 67 GO terms that were significantly overrepresented ($p_{perm} < 0.05$) and represented by more than one gene (Supplementary Data 6). Notably, there were four GO terms related to metabolism: fatty acid derivative biosynthetic process (GO: 1901570, $p_{perm} = 0.001$), fatty acid derivative metabolic process (GO: 1901568, $p_{perm} = 0.001$), regulation of carbohydrate metabolic process (GO: 0006109, $p_{perm} = 0.001$), and regulation of

carbohydrate catabolic process (GO: 0043470, $p_{perm}$ = 0.01). These functions may be related to diet change of dingoes. The candidate genes include also genes related to, e.g., reproduction and neuronal function which may have played roles in the feralization adaptation of dingoes: *Prss37* (Protease, Serine 37), shown to be required for male fertility in mice[55], *ARHGEF7* (Rho Guanine Nucleotide Exchange Factor 7) which promotes the formation of neural spine and synapses in hippocampal neurons[56], and *TAS2R5* (Taste 2 Receptor Member 5) which plays a role in the perception of bitterness[57]. Interestingly, four of the genes have previously been indicated to be related to the domestication of dogs: *SLC5A1* and *ZNF516* were identified by Axelsson et al.[20], *TAS2R5* and *ZNF516* by Cagan et al.[58], and a novel gene (ENSCAFG00000023577) was found by Pendleton et al.[22].

**Change in the candidate regions in two processes.** To compare the candidate regions in the domestication and feralization steps we also performed the PBS analysis using the formula PBS2 = $\frac{T_{ds} + T_{ws} - T_{dw}}{2}$ (Eq. 2, see Methods for details), comparing dingoes, dogs from Southern East Asia and wolves, to identify genomic regions in dingoes that were more similar to wolf than to dog. High PBS2 values (the first percentile rank was used as threshold, 0.0766) indicate large difference between dog and dingo and between dog and wolf, but low difference between wolf and dingo. This identifies regions with large difference between dingo/wolf and dog, while regions with smaller difference may be ignored. Based on this, we identified 1100 windows with high PBS2 values, and compared these windows with the 87 windows identified as candidates associated with feralization by PBS1 and iHS. This identified 17 overlapping windows, containing 13 genes, with high values for PBS1 and iHS as well as for PBS2 (Table 1, Fig. 4). This suggests that these 13 genes were under positive selection in dingoes and also more similar to gray wolves than to domestic dogs. Functional annotation showed that four of these 13 genes are associated with neurodevelopment, metabolism and reproduction (Table 1, Supplementary Table 4). Specifically, *ARHGEF7* (Rho Guanine Nucleotide Exchange Factor 7) may promote the formation of neural spine and synapses in hippocampal neurons[56], *SLC5A1* plays an important role in the absorption of glucose and sodium[59], *TAS2R5* (Taste 2 Receptor Member 5) may play a role in the perception of bitterness[57] and *Prss37* (Protease, Serine 37) is related to reproduction[55]. We visualized the genotypes in 70 canine samples for these four genes. This showed that the genotypes for dingo and NGSD were almost identical for all four genes, and that dingo/NGSD had low diversity with most positions being homozygote for the non-reference variants (Supplementary Fig. 12). It also showed that dingo/NGSD were more similar to the wolves than to dogs, dingo/NGSD and wolves sharing the non-reference homozygote type in many positions. The dogs were more heterogenous, with a large proportion of

**Table 1 The 13 genes under selection in both domestication and feralization.**

| Window coordinates | Gene |
|---|---|
| chr10:13800001-13820001 | TRHDE |
| chr10:17020001-17060001 | TUBGCP6, SELENOO, ENSCAFG00000000697, TRABD, PANX2 |
| chr16:7340001-7360001 | PRSS37, ENSCAFG00000003879 |
| chr16:7400001-7420001 | RF00026, TAS2R5 |
| chr22:59220001-59240001 | ARHGEF7 |
| chr23:26980001-27000001 | ANKRD28 |
| chr26:24940001-24960001 | SLC5A1 |

sites that were heterozygous, or homozygous for the reference variants. However, there was a large difference between the dogs from Southern East Asia and Europe. For all four genes, diversity was largest for the dogs from Southern East Asia, which had all three genotype variants in most positions. In contrast, for three of the genes, *SLC5A1*, *TAS2R5*, and *Prss37*, the European breed dogs were homozygous for the reference variant across almost the whole region. This suggests that selection has occurred in European breeds but not in the dogs from Southern East Asia, which would imply that these genes were not under selection during the domestication of the dog but during the later development of the European breeds. It also suggests that selection in these genes had not occurred for the ancestors of dingoes but occurred during the feralization of dingoes. Given the strong bottleneck in dingoes, we also performed simulations based on the inferred demographic history as null expectation for selection, with consistent results (Supplementary Figs. 13–15).

**Functional assay revealed that a mutation in dingo gives decreased enhancer activity for *ARHGEF7*.** *ARHGEF7* is related to neural function[56] and may therefore be involved in behavior changes in the development from dog to dingo. We found an A-to-G mutation (chr 22: 59234593) within the *ARHGEF7* gene, which had a very high allele frequency in dingoes (100%) and wolves (93.3%) compared to indigenous dogs from southern East Asia (32.5%). Furthermore, detailed bioinformatics analysis showed that the A-to-G mutation may influence the expression of *ARHGEF7* since it is located in a transcription factor-binding site[60]. To test whether the SNP variants can actually affect expression of *ARHGEF7*, we performed dual-luciferase reporter experiments (enhancer assay using pGL3-promoter vectors) on two human cell lines (Daoy, human medullablastoma and HEK293, human embryonic kidney) and one canine cell line (MDCK, Madin-Darby Canine kidney). The analyses showed that all three cell lines displayed significantly lower enhancer activities for SNP-G than for SNP-A (Fig. 5), suggesting that SNP-G may confer decreased endogenous *ARHGEF7* production.

**Discussion**
In this study, we have investigated the process of feralization on the genomic level, using the dingo as a model. The analyses of population structure and demography reinforces that the dingo is an excellent model for this, because its feralization started 8000 years ago and because it has then remained isolated from its domestic and wild ancestors until the last 200 years. This makes the dingo a unique tool for identifying genomic regions under positive selection in the feralization process without confusing the impact of feralization with hybridization to ancestral populations.

Our study has presented important new findings about the origins and history of the dingo. In the past decades, numerous population genetic studies of the dingo have been performed based on mitochondrial and Y-chromosomal DNA[24,39,42,48], indicating an origin from East Asian domestic dogs but lacking in precision about timing, routes of arrival to Australia and demographics. Our studies of whole genomes in dingoes and related canids clarify several of these details. Our analyses of phylogeny, population structure, and demography as well as selection analysis show that the dingo is a genetically distinct population clearly differentiated from the domestic dog. The selection analyses indicate that 8000 years of feralization has affected numerous genes linked to, e.g., neurodevelopment, metabolism and reproduction.

TreeMix, phylogenetic analyses, and outgroup f3 analyses all identified indigenous dogs from southern China and Indonesian village dogs as the dog populations which are genetically most

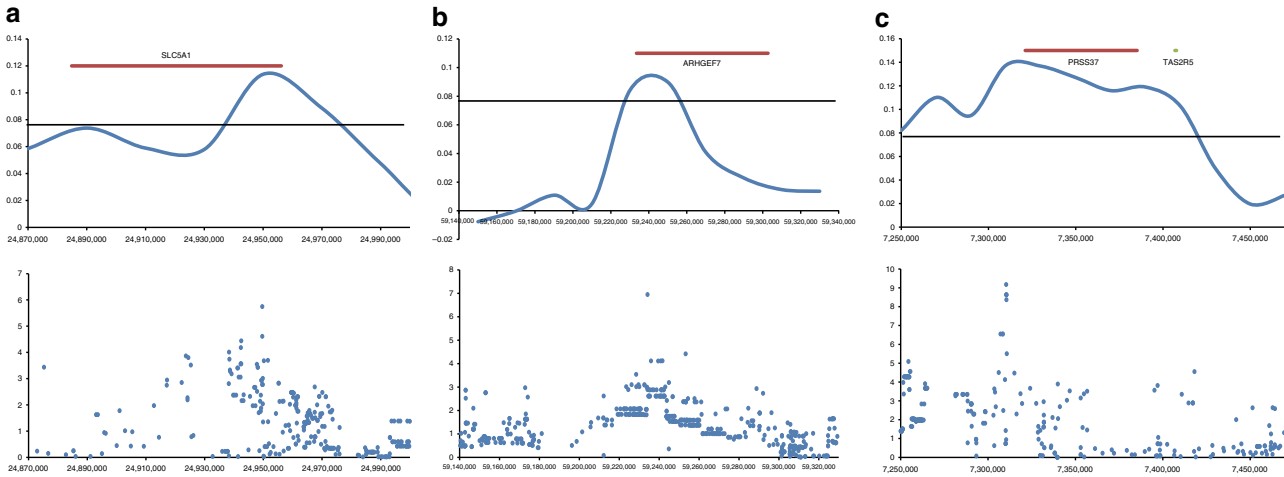

**Fig. 4 The result of iHS and PBS2 for three candidate regions.** Blue curves indicate PBS2 value, blue dots indicate iHS value. Thick red horizontal line indicates range of gene, and thin black horizontal line gives the threshold.

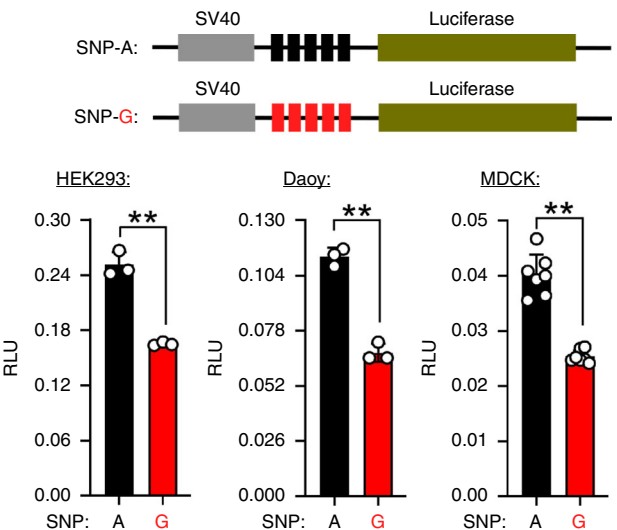

**Fig. 5 Functional assay of ARHGEF7.** Dual-luciferase reporter experiments (enhancer assay using pGL3-promoter vectors) using two different human cell lines (HEK293, human embryonic kidney and Daoy, human medullablastoma) and one canine cell line (MDCK, Madin-Darby Canine kidney). All experiments were replicated at least three times. Data are expressed by mean ± SEM with the corresponding data points. Graphpad software was used for statistical analysis. Comparisons were performed using two-tails Student's t test. **represents $p < 0.01$.

closely related to dingoes. The genomic data provides strong evidence that the dingo originates from domesticated dogs in southern East Asia, which migrated via Island Southeast Asia 9900 years ago, to eventually reach Australia 8300 years ago, and the mitochondrial data supports this picture. With this data, we can reject two previous hypotheses about the origin and migration routes of dingoes. Based on similarity in skeletal anatomy to Indian pariah dogs and wolves, and gene flow from ancient Indian populations to indigenous Australians dated at BP4200[61], it has been suggested that the dingo ancestors came from India[25,40]. An alternative theory has been that dingoes originate from dogs introduced with the Austronesian expansion into Island Southeast Asia, which arrived in New Guinea about 3600 years ago[24]. However, the genomic analyses, as well as previous mtDNA data, clearly indicate an origin from dogs in southern East Asia, which arrived to Australia via mainland Southeast Asia,

and our demographic analysis indicates an arrival in Australia 8300 years ago, well before the Austronesian expansion[62–64]. Thus, the genetic data clearly suggest that the dingoes originate from domestic dogs in southern East Asia that migrated via mainland and Island Southeast Asia to reach Australia 8300 years ago, but the human population that was involved in this migration remains unknown.

The results showed that dingoes and NGSDs are genetically very closely related, indicating a common origin from dogs in Island Southeast Asia around 8300 years ago. We also note that there is a phylogeographic structure in the dingo population recorded by nuclear as well as mitochondrial data. This was possibly caused by drift in the earliest formation of the dingo population, but may also relate to an origin from more than one introduction to Australia, but if so from a very homogenous source population and at similar points of time.

Our analyses identify 50 candidate genes in genomic regions under selection in dingo, and find an overrepresentation of genes correlated in particular to digestion, metabolism and reproduction. This indicates an adaptation to a new environment, in the form of a change of diet and changed sexual and reproductive mechanisms. This agrees with the two previous studies of genomic change under feralization, on feral chicken and rice. In the feral rice, genomic regions containing numerous genes correlated with adaptation to the new environment were identified, linked to, e.g., flowering time, reproduction and stress response[15]. In the feral chickens, especially genes correlated with sexual selection and reproduction were identified, e.g., genes correlated with fecundity traits, which may be targets of selection that facilitated the feralization[10]. It is notable that genes correlated with sexual selection and reproduction were identified in feral chicken and rice as well as in the dingo, indicating that change in reproduction mechanisms is an important effect of feralization in both animals and plants.

There is considerable difference in diet between domestic dogs and the two related wild canids, wolves and dingoes. The wild canids have a diet consisting predominantly of meat, while domestic dogs normally eat considerable amounts of vegetable food, provided by humans[21,33,44,65,66]. This diet change has been shown to be reflected by strong selection for improved digestion of starch in domestic dogs[44]. This is manifested most prominently by expansion of copy numbers of the gene for pancreatic amylase (AMY2B) in most dogs, but dingoes have the non-expanded wild type found in wolves[67,68]. In our selection analyses we now also found feralization genes related to digestion and

absorption. This indicates that diet change has implied a major environmental influence on the dingo, resulting in genomic change.

We demonstrate that the feralization of the dingo induced positive selection on genomic regions correlated to neurodevelopment, metabolism and reproduction. We also compared the genomic regions under selection in the feralization step to those selected in the domestication step, by comparing PBS2 among dingoes, dogs from Southern East Asia and wolves, where a high PBS2 indicate genomic regions in dingoes that were more similar to wolf than to dog. This analysis identified 13 genes in regions under selection in dingoes which were also more similar to gray wolves than to domestic dogs. This may indicate that selection on the 13 genes occurred in the dog lineage after the split from the dingo ancestors. However, inspection of the genotypes for four of these genes suggests that selection did not occur during the domestication of the dog, but during the later development of the European breeds and during the feralization of dingoes.

Importantly, two of the 13 genes are related to neurodevelopment (ARHGEF7 and PANX2), and therefore possibly involved in behavior change necessary for feralization. We performed a functional analysis on one of these genes, ARHGEF7 which promotes the formation of spines and synapses in hippocampal neurons[56]. This test showed that a SNP found in dingo, located in a transcription factor-binding site, gives significantly lower enhancer activities. Hippocampus plays important roles in response inhibition, memory, and spatial cognition[69,70], and some studies suggest that hippocampus relates to purposive behaviorism[71]. Therefore, changes in expression of this gene may be related to behavior changes in the dingo, linked to the adaptations to a wild environment.

In this study, we show that the feralization of the dingo induced positive selection on genomic regions correlated to neurodevelopment, metabolism and reproduction. We demonstrate that an SNP variant for one of these genes found in dingo gives significantly decreased enhancer activities. We also establish that the dingo originated around 8300 years ago from domestic dogs in southern East Asia. The dingo has thereafter remained isolated, and under 8300 years of adaptation to a life in the wild it has developed into a genetically distinct population clearly differentiated from its domestic ancestors.

## Methods

**Samples and sequences**. We examined whole-genome sequences from the largest and most diverse group of dingo studied to date, amassing a dataset of 109 canines around the world. The map was drawn by the R Packages (maps: https://CRAN.R-project.org/package=maps). We sequenced genomes of 10 dingoes and 2 New Guinea Singing Dogs in the study. Total genomic DNA was extracted from blood or tissue samples using the phenol/chloroform method. For each individual, 1–3 μg of DNA was sheared into fragments of 200–800 bp with the Covaris system. DNA fragments were then processed and sequenced using the Illumina HiSeq 2000 platform. Raw sequence reads were mapped to the dog reference genome (Canfam3)[72] using the bwa mem –M (version 0.7.10-r789)[35]. We used PICARD (version 1.87) to remove duplicated reads and merged BAM files for multiple lanes. Sequences were then locally realigned and base-recalibrated using the Genome Analysis Tool Kit (GATK, version 2.5-2-gf57256b)[36]. Base quality was recalibrated using GATK BQSR to produce final BAM files. Sequence data were next subjected to a strategic procedure for variant calling using the UnifiedGenotypeCaller in Genome Analysis Tool Kit (GATK). Raw variants were then recalibrated using the Variant Quality Score Recalibration (VQSR). During base and variant recalibration, a list of known SNPs downloaded from the Ensembl database (ftp://ftp.ensembl.org/pub/release-73/variation/vcf/canis_familiaris/) was used as the training set.

**Genetic diversity and population structure**. Genetic diversity was calculated using VCFtools[73]. Principal component analysis was made using the smartPCA[74]. After thinning to a single SNP per 50 kb window, population structure analysis was performed using the block relaxation algorithm implemented in the ADMIXTURE software[37] The NJ phylogenetic tree was built by MEGA7[75]. The ML phylogenetic tree was built by RAxML-8.0.12[76]. We also used TreeMix[77] to investigate the genetic relationship and population level admixture.

**D-statistics and outgroup-f3 analysis**. We performed D-statistics analysis using qpDstat in the Admixtools[38] software package to test events of gene flow between the dingoes and European breeds in the form of D (Dhole, European breed; Pop1, Pop2). We performed three tests, where (i) Pop1 was all dog groups tested in turn and Pop2 was each individual dingo, (ii) Pop1 was all dog groups and the dingoes tested in turn and Pop2 was the NGSDs, and (iii) Pop1 was all dingo individuals tested in turn and Pop2 was all other dingo individuals tested in turn. A significantly positive D value (>3) suggests that Pop2 shows higher affinity to European breed than do Pop1, and that there may be admixture between European breed and Pop2. If instead D is significantly negative (<−3), there may be admixture between European breed and Pop1. The qp3pop program[38] in the Admixtools[38] software package was used to perform outgroup f3-statistics analysis in the form of f3(Dingoes, Pop2; Dhole)[38,41], to assess the relative genetic similarity of the dingo population to the other dogs, where high f3 values indicate a high degree of shared genetic history between the populations[37].

**Population history**. We inferred a complete demographic model for dingo and other dogs, including population divergence times and population size using the Generalized Phylogenetic Coalescent Sampler (G-PhoCS)[43]. We dated two important internal nodes in the history of dingo: the divergence time between indigenous dogs from southern East Asia and Indonesian village dogs (Tau1), and the divergence time between Indonesian village dogs and dingo (Tau2). The phylogeny input was (indigenous dogs from southern East Asia, (Indonesian village dogs, dingoes)). We made two rounds of G-PhoCS analysis (Supplementary Table 2, Supplementary Data 3), and in each analysis we repeated the computations three times by randomly picking 1000 neutral loci (Supplementary Table 3, Supplementary Data 4), and took the average as the result. For the first round, we randomly selected 3 samples from all dingoes and for the second round, we replaced the D05 to D08 and D06 to D01 respectively, since the random procedure did not involve D05 and D06. The complete demographic history was inferred for dingoes, including population divergence times, ancestral population size, and migration rates based on the 1000 neutral loci. The parameters were inferred in a Bayesian manner using a Markov Chain Monte Carlo (MCMC) to jointly sample model parameters and genealogies of the input loci. Burn-in and convergence of each run were determined with TRACER 1.5[78]. For the control file of G-PhoCS, divergence times in units of years, effective population sizes, and migration rates were calibrated by the estimates of generation time and neutral mutation rate from previous studies. A generation time of 3 years, a neutral mutation rate of 1.3e−09 per site per year were used to convert the population sizes and scaled time into real sizes and time. The mutation rate of $1.3 \times 10^{-9}$ is calibrated by ancient DNA and used in many previous studies[22,40,41], and the result when we use this as the mutation rate is in agreement with the mitochondrial genome estimate.

**Mitochondrial genome analysis**. The NJ phylogenetic tree was built using MEGA[75]. Bayesian analysis was made using Beast[79], assuming a mutation rate of $7.7 \times 10^{-8}$ per site per year with SD $5.48 \times 10$ according to Thalmann et al.[49] Burn-in and convergence of each run were determined with TRACER 1.5[78].

**Selection analysis**. We performed the PBS statistics using the following formula:

$$PBS1 = \frac{T_{ds} + T_{db} - T_{bs}}{2} \qquad (1)$$

and

$$PBS2 = \frac{T_{ds} + T_{ws} - T_{dw}}{2} \qquad (2)$$

where $T$ is computed by

$$T = -\log(1 - Fst) \qquad (3)$$

and $T_{ds}$ computed from Fst between dingoes and dogs from SE Asia/South China, $T_{db}$ computed from Fst between dingoes and European breeds, $T_{bs}$ computed from Fst between European breeds and dogs from SE Asia/South China, $T_{dw}$ computed from Fst between Gray wolves and dingoes, and $T_{ws}$ computed from Fst between Gray wolves and dogs from SE Asia/South China.

We phased our data using the software SHAPEIT[80] based on the genetic recombination map from Auton et al.[33]. We calculated iHS in dingo using the software of selscan[81], and normalized the scores by norm(in the software of selscan) with a 20 kb sliding window across the autosomes. We identified windows as candidate regions for selection if 30% of sites within them had an iHS absolute value above the threshold (top1% of iHS). Simulations were performed using the ms program[82]. We simulated five groups of genome sequences (wolves, indigenous dogs from southern China, Indonesian indigenous dogs, and dingoes for PBS2 and iHS; indigenous dogs from southern China, European breeds, Indonesian indigenous dogs, and dingoes for PBS1) under a neutral evolutionary model considering the inferred demographic history. A mutation rate of $2.2 \times 10^{-9}$ per site per year with a generation time of 3 years was assumed. Since our demographic analysis did not include wolves and European breeds, the effective population sizes and split time for wolves and European breeds were taken from Wang et al.[3] and Liu et al.[83].

Scripts used for simulation were as follows:
For PBS1:
ms 90 110000 -t 52.5 -r 80 20000 -I 4 42 20 6 22 -n 1 0.17917 -n 2 0.069 -n 3 0.19305 -n 4 0.01256 -ej 0.00409 4 3 -en 0.00409 3 0.09217 -ej 0.00488 3 1 -ej 0.01383 2 1 -em 0.00409 3 1 5.236
For PBS2:
ms 112 110000 -t 52.5 -r 80 20000 -I 4 42 42 6 22 -n 1 0.87120 -n 2 0.17917 -n 3 0.19305 -n 4 0.01256 -ej 0.00409 4 3 -en 0.00409 3 0.09217 -ej 0.00488 3 2 -en 0.00488 2 0.56680 -ej 0.0275 2 1 -em 0.00409 3 2 5.236
For iHS:
ms 112 11000 -t 52.5 -r 80 20000 -I 4 42 42 6 22 -n 1 0.87120 -n 2 0.17917 -n 3 0.19305 -n 4 0.01256 -ej 0.00409 4 3 -en 0.00409 3 0.09217 -ej 0.00488 3 2 -en 0.00488 2 0.56680 -ej 0.0275 2 1 -em 0.00409 3 2 5.236

**Gene enrichment analysis**. We randomly select 87 windows with 20Kb in the whole genome defined as the permuted gene sets. GO analyses were performed on both observed and permuted gene sets using the parent-child model[53] in the topGO R package[54]. Permutation-based $p$ values (pperm)[22] were produced for all GO terms by comparing the observed parent-child test score with the results of the 1000 permutations using the formula pperm $= (Xperm + 1)/(N + 1)$, where Xperm is the number of instances where a permutation obtained a parent-child $p$ value less than or equal to the observed $p$ value, and $N$ is 1000. GO terms with pperm values less than 0.05. And we download go annotation sets from NCBI (https://ftp.ncbi. nlm.nih.gov/gene/DATA/gene2go.gz).

**Functional test using dual-luciferase reporter assay**. To construct *ARHGEF7* enhancer SNP reporters, we inserted five repeats of the 50 bp fragments arounding the indicated SNP site into the pGL3-Promoter vector (Promega) within the MluI and XhoI sites. We verified all recombinant clones by sequencing. Daoy (human medullablastoma), HEK293 (human embryonic kidney) and MDCK (Madin-Darby Canine Kidney) cells were cultured in high-glucose Dulbecco's modified Eagle's mediun (DMEM) (Corning) with 10% fetal bovine serum (FBS) (Gibco) at 37 °C in 5% $CO_2$ condition. For luciferase reporter assays, Daoy, HEK293 and MDCK cells were transfected with the indicated reporter plasmids together with the same TK-Renilla internal control reporter vectors by using the lipofectamine 2000 transfection reagent (Invitrogen) and changed with the fresh medium at 6 h after transfection. According to the manufacture's instruction, luciferase activity was measured at 36 h after transfection by using the Dual-Luciferase Reporter Assay System (Promega). All assays were performed in at least three independent experiments with a minimum of three replicates.

**Ethics statement**. We have complied with all relevant ethical regulations for animal testing and research. Australian Government Export permit number N39585 and University of New South Wales Ethic's Approval 16/77B to Professor Bill Ballard.

**Reporting summary**. Further information on research design is available in the Nature Research Reporting Summary linked to this article.

## Data availability
The raw sequence data from this study have been submitted to the GSA (http://gsa.big.ac.cn/) under accession CRA000200 for raw data of genomes. This project has also been deposited at the National Center for Biotechnology Information (NCBI) Sequence Read Archive database with the accession code PRJNA593363 (SRP234866). We have used downloaded data from published articles: SRA307300[3], SRP044399[32], SRP035294[31], SRP062184[21], SRP062060[33] and SRP058219[30]. The dog reference genome is Canfam3[72]. The source data underlying Figs. 1b, c, 4a–c, 5 and Supplementary Fig. 9 and Supplementary Table 4 are provided as a Source Data file.

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

## Acknowledgements

The authors thank J. William O. Ballard of the University of New South Wales for providing dingo samples and Janice Koler-Matznick for providing NGSD samples. This work was supported by grants from the NSFC (91531303, 31571353 and 91731304), the Breakthrough Project of Strategic Priority Program of the Chinese Academy of Sciences (CAS) (XDB13000000), Carl Trygger's Stiftelse and Agria and SKK Forskningsfond. G.D.W. is supported by the Youth Innovation Promotion Association of CAS and the 13th Five-year Informatization Plan of CAS (Grant No. XXH13503-05). L.L.Z. was sponsored by the China Scholarship Council (CSC#201700260248).

## Author contributions

Y.P.Z., P.S. and B.Y.M. supervised the research. S.J.Z. and G.D.W. designed the research. S.J.Z. and G.D.W. performed the research and analyzed data. P.M. performed Functional assay. L.L.Z performed mtDNA analyses. S.J.Z., T.T.Y. and L.W. carried out data submission. Y.H.L. and Y.W. helped perform the analysis with constructive discussions. M.W. and Y.P.M. performed experiments. S.J.Z. and G.D.W. wrote the manuscript. N.O. revised the manuscript. Y.P.Z., P.S. and B.Y.M. approved the final version.

## Competing interests

The authors declare no competing interests.
