## [Peer Review File · Nature Communications]

Reviewers' Comments:

Reviewer #1:

None

Reviewer #2:

Remarks to the Author:

The authors present an interesting genomic analysis of sequence data from dingoes in the context of other dogs and wolves. They reconstruct history of the dingo group and perform some analysis of candidate selection regions, arguing that after feralization regions of the genome associated with domestication in dingos arguing that selection during domestication was reversed during feralization. Overall the manuscript is clearly written and reports several interesting findings. However, key details are lacking, the demographic analysis is somewhat simplistic and does not make full use of available data. The selection arguments are also somewhat superficial. The summary stats used are not calibrated based on expectations of a neutral model (which the authors are in a very strong position to do, given the demographic inference work in the first part of the manuscript). It is not clear to me that the comparisons across wolf, dingo, dog allow for detection of true domestication or feralization signals. It seems possible as well that rather than "reversed" selection, there was additional selection in dogs (after the dog-dingo split) that gives rise to these signals. Additionally, it isn't clear to me if the author's model is that the 'selected' alleles in domestication were not fully fixed, or if new mutations arose in dingos, or if there was somehow gene flow from wolves. Overall a much more detailed analyses and description of the approach and the assumptions is required. I feel that additional analysis and clarification could result in a substantially strengthened manuscript with more robust conclusions.

Additional specific comments are given below.

Overall, key details seem lacking from the methods.

Pg 4 " of the alpha-amylase locus giving improved starch digestion in domestic dogs" This could be reworded. The amylase duplication is not found in all modern domestic dogs and several analyses have shown it is not a variant associated with domestication. Thus, in this locus dingo are not unique relative domestic dogs.

It would better to supplement PCA and NJ trees with methods such as F3 or F4 statistics to more formally show which group dingos are most closely related to. Since there are multiple samples, an approach such as Tree-Mix, which is based on models of drift rather than simple allele sharing as in NJ trees, may be more appropriate.

Also, in the NJ tree in Figure 1 there is not metric of confidence in the tree structure. A block bootstrap should be employed. Also, it would be informative to label individual samples.

What method was used for the structure analysis? ADMIXTURE? This needs to be referenced appropriately. The choice of K should be discussed as well. If ADMIXTURE was used, what was the cross-validation rate?

Two of the dingo samples appear to have substantial gene flow from dogs. It is not clear if these samples are included in the subsequent analysis (they are excluded from the NJ tree). Also, it is not clear if the apparent dingo populations are treated as one or modeled separately, or what effect this would have on the various conclusions.

Diversity of dingos (supp fig 4). It is not clear what the parameter "theta pi" is. The sample collection is very skewed as for example quite distinct wolf groups are considered as one group even though they split long ago. This statistic and the meaning of this analysis should be explained.

The G-PHOCS analysis should also consider effects of gene flow on the estimates, as they split times are likely to be very sensitive to this. The 2.2×10^{-9} site/year mutation rate also does not seem appropriate, with studies calibrating using ancient DNA seem to suggest $\sim 1.3 \times 10^{-9}$ per year. The effect of the assumption of mutation rate should be discussed in detail since the dating of the dingo split is a key part of the study.

It is also not clear which samples were used in the G-PHOCS analysis. All of the dingos together? Excluding the two with large admixture? Which samples for the other parts? How were the variants called for GPHOCS analysis? Just using the regular GATK VCF after VQSR? Overall the description of this analysis is lacking.

The selection analysis is also somewhat superficial. Outlier approaches based on statistics are used, but there is not any way to set expectations. Since the authors infer the demographic history, these should be utilized to set null expectations for selection, particularly given the apparent strong bottleneck in dingos. Such simulations are critical for interpreting the results.

I think the GO enrichment also has several shortcomings. First, it seems that there is no correction for multiple hypothesis testing. It is also not clear which method was used for GO enrichment testing. Furthermore, these methods usually do not account for spatial structure of genes along chromosomes, gene size etc, and are not directly appropriate for genome based enrichment analysis.

More importantly, I do not think that the core claim of "reversed selection" is well supported by the data as presented. Could not the described F_{st} pattern simply reflect pressure in modern dogs that occurred after the dog-dingo split, rather than the 'reverse selection' argument made by the authors. Haplotype information should be very informative for such a scenario, in addition to the simple F_{st} in windows approach.

Relatedly, many studies have attempted to find genome regions associated with dog domestication. For example, Pendleton et al recently used village dogs and ancient DNA to try to identify selection during domestication vs selection during later dog evolution. It would be valuable to compare the purported 'reversed selection' signals with the findings of that study.

Reviewer #3:

Remarks to the Author:

The authors present a study of the genomic mechanism of feralization, using the dingo, a canid species native to Australia, as a model in this manuscript. The manuscript presents novel insights into the feralization process, and presents

good evidence for the role of certain behavior related genes in the process.

The article suggests that Indonesia, possibly through SE Asia, is the ancestral population of the dingos. Further, they suggest the presence of "reverse" selection on some of the genes they find to be outliers in F_{st} scans. These genes, according

to the authors, underwent a second round of selection in the feralization process, to bestow upon the dingos traits that resemble those of wolves much more than the same traits in dogs.

The methods and statistics used in these analyses need

a closer look, as specific statistics (such as PBS and LSD) exist, exactly for the scenarios that the

authors are examining here. I would highly recommend using these statistics instead of the adhoc F_{st} based comparisons used here.

The functional verification of one of the putative genes under "reverse" selection is indeed a very nice part of the article, and kudos to the authors for following up on one of the most interesting genes of their set of findings.

In terms of the scientific impact of the manuscript, the work presented here clarifies the geographic origin of the dingoes. The authors propose SE Asia as the source, with migration through the SE Asian islands, finally to colonize Australia. They dismiss the previous claims of an Austronesian human expansion as a possible source, based on some of the population history and demography dates. These dates are based on mutation rate estimates that are still not definitively agreed upon in the canid research community. Thus, in my opinion, the Austronesian expansion could still be a viable explanation.

Overall, the manuscript is well written and presented, but could definitely undergo a round of proof reading and copy editing, to improve some of the phrasing. There are some places where the sentences are a bit confusing.

Specific comments:

Page 5: "After strict filtering, we identified ~24.7 million autosomal SNPs for further analysis." Insufficient details provided here. Please provide details on SNP calling and filtering. It is impossible to replicate these findings (or comment on their robustness without details). This is a recurrent theme, and the authors would do well to share more details on their analyses.

Page 5: "In a two-dimensional plot of the genotypes, there is a clear separation in four groups: Tibetan wolves, other wolves, dogs and dingoes/NGSDs." In figure 1B, I see 3 of the four groups mentioned wolves, dogs and dingoes/NGSD. The lone Tibetan wolf clusters with the dogs - a bit of a surprise but can be explained by the low sample size (1 sample of Tibetan wolf), and the known oddities in the genetic history of the Tibetan wolf. I would suggest that the authors focus on the three groups where there is a clear signal of separation.

Page 5/6: "Among the dogs, Indonesian village dogs cluster closest to the dingoes/NGSDs, followed by indigenous dogs from southern East Asia (Fig. 1B). Notably, India and Taiwan have been suggested as possible origins for the dingo, but the dogs from these regions cluster relatively far from the dingoes." I suggest that the authors refrain from using the PCA and other dimension reduction representations to indicate origin of the dingo. They have much better and well supported evidence in both admixture and neighbor joining trees. First, the amount of variance explained by PC2 is not very high. Second, both the dog cluster and the dingo cluster are spread out on the PC2 axis, suggesting some level of genetic heterogeneity in these clusters, and therefore, the closeness of the Indonesian dogs might have resulted from gene flow in the (early) history of the dingoes, and especially the NGSD. I said earlier, better evidence is provided by some of their other analyses.

Page 6: "The results indicate gene flow between the two dingoes (D05 and D06) and European breeds ($D=0.1027$ and 0.0827 , $Z = 14.551$ and 10.221) (Supplementary table 2)." The authors present convincing evidence that there is gene flow between D05, D06 and European breeds, but I would like to see the results of the same D statistic test performed on the other dingoes as well (in supplementary table 2), if for nothing else, then just for comparative purposes.

Page 7: The parameter " θ_n " should have the "n" in subscript. "This suggests a severe bottleneck event in the evolutionary history of dingoes, which may explain the long phylogenetic branch." Although a bottleneck is one possible explanation, there are other explanations that the authors should

consider. Long periods of isolation with low effective population size, as was probably the case in the dingos, will lead to long external branches and low diversity. This is very reasonable alternative explanation to the severe bottleneck hypothesis.

Page 9: As shown by the authors and previous studies, there is clear evidence of geographic structure in the dingos. But there is little to no evidence of two introductions of the dingo into Australia, and the authors provide none, and do not test this hypothesis after presenting it.

Page 10: First paragraph. Needs some rewriting to clarify, esp. the sentence starting with "A window ...".

Page 10: Use of F_{st} between the dingos and the Indonesian/South East Asian dogs is a valid way to identify genomic regions that differ between these two groups, but this does not allow identifying these regions are being positively selected in specifically the dingo lineage. These genes might have undergone changes in the other lineage as well. Using iHS to add support to these windows does alleviate these issues, but it would be good to provide the distribution of iHS window scores in the supplementary material.

Page 10: Section "Genes with reversible selection in domestication and feralization" This section uses 3 different pairwise F_{st} measures to figure out the branch on which the selection might have occurred. A much more principled approach, which should be used here, is the Population Branch Statistic (PBS) which is designed to do exactly what the authors are trying to achieve here. I would highly recommend switching to the PBS statistic instead of this adhoc measure using three different F_{st} statistic. This would allow them to obtain an empirical distribution for the PBS statistic which then can be used to select outlier regions of the genome.

Page 11: "This suggests that these 14 genes evolved during both the feralization and the domestication steps, such that feralization imposed reversed selection compared to the domestication." The authors argue that the signature of high F_{st} between wolves and dogs, and low F_{st} between dingos and wolves shows that these genes underwent a selection in the reverse direction (to revert back to the wolf form) in the dingos. This is not the most parsimonious explanation for this observation. One possible explanation would be the selection on these genes in the South East Asian dogs after the split of the dingos. The authors propose that this split happened ~5000 years ago, giving plenty of time for this selection to have happened in the SE Asian dogs. The use of the PBS statistic would address these problems.

Page 13: "Interestingly, the demographic analysis indicates that the dingo ancestors originated from southern China 5,300-6,600 years ago, at which time large-scale rice farming was established in this region."

The authors do provide evidence that Indonesian dogs are the ancestors of dingos, but this statement here suggests that Southern China was the ancestral population of the dingos, which is only marginally (at least without further analyses) supported in this manuscript.

The lack of the *AMY2B* copy number increase in the dingos also suggests that these split from the SE Asian dogs before the copy number expansion in these dogs. If I am not mistaken, this is the point that the authors are trying to make in the last sentence of the paragraph. If so, I would suggest a rewrite to explicitly state this, since the connection to rice farming is currently very tenuous.

Supplementary figure 2: Please provide bootstrap/local posterior probability support values for the nodes in this tree.

Reviewer #1 (Remarks to the Author):

The authors present an interesting genomic analysis of sequence data from dingoes in the context of other dogs and wolves. They reconstruct history of the dingo group and perform some analysis of candidate selection regions, arguing that after feralization regions of the genome associated with domestication in dingoes arguing that selection during domestication was reversed during feralization. Overall the manuscript is clearly written and reports several interesting findings. However, key details are lacking, the demographic analysis is somewhat simplistic and does not make full use of available data. The selection arguments are also somewhat superficial. The summary stats used are not calibrated based on expectations of a neutral model (which the authors are in a very strong position to do, given the demographic inference work in the first part of the manuscript). It is not clear to me that the comparisons across wolf, dingo, dog allow for detection of true domestication or feralization signals. It seems possible as well that rather than “reversed” selection, there was additional selection in dogs (after the dog-dingo split) that gives rise to these signals. Additionally, it isn’t clear to me if the author’s model is that the ‘selected’ alleles in domestication were not fully fixed, or if new mutations arose in dingoes, or if there was somehow gene flow from wolves. Overall a much more detailed analyses and description of the approach and the assumptions is required. I feel that additional analysis and clarification could result in a substantially strengthened manuscript with more robust conclusions.

Reply and revision: Thank you for your valuable comments. We have added details in the Materials and Methods, refined the demographic analysis, and performed the PBS and haplotype analysis to solve the issue of reversible selection. We also performed the simulation analysis as null hypothesis to refine the selection. We added the analysis of the outgroup f3 and Treemix. Based on the result of Treemix and Australia’s position outside the natural range of wolves, there was no gene flow from wolves to dingoes. We have revised the manuscript accordingly.

Additional specific comments are given below.

Overall, key details seem lacking from the methods.

Reply: We have revised the Materials and Methods accordingly.

Pg 4 “ of the alpha-amylase locus giving improved starch digestion in domestic dogs” This could be reworded. The amylase duplication is not found in all modern domestic dogs and several analyses have shown it is not a variant associated with domestication. Thus, in this locus dingo are not unique relative domestic dogs.

Reply and revision: Our claim was not rigorous, and we have revised and discussed it accordingly.

It would better to supplement PCA and NJ trees with methods such as F3 or F4 statistics to more formally show which group dingoes are most closely related to. Since there are multiple samples, an approach such as Tree-Mix, which is based on models of drift rather than simple allele sharing as in NJ trees, may be more appropriate.

Reply and revision: We have added these analyses. We performed the outgroup F3 and the result revealed that the NGSDs are most closely related to dingoes, and then Indonesia dog, then indigenous dogs from southern East Asia. Please see the results in detail in supplementary table 2. We also performed Treemix, and the topology is consistent with the aforesaid phylogeny constructed by the NJ approach. Based on Treemix, we just find one candidate gene flow from the dingo/NGSD to the lineage of Papua New Guinea village dogs. We revised the whole manuscript accordingly.

Also, in the NJ tree in Figure 1 there is not metric of confidence in the tree structure. A block bootstrap should be employed. Also, it would be informative to label individual samples.

Reply and revision: Considering the image resolution, we do not put all bootstraps and labels in Figure 1, but in the Supplementary Figure S4, S5. Thus, we added the bootstraps of the main nodes, and the labels of the population in Figure 1.

What method was used for the structure analysis? ADMIXTURE? This needs to be referenced appropriately. The choice of K should be discussed as well. If ADMIXTURE was used, what was the cross-validation rate?

Reply and revision: We have revised it accordingly.

Two of the dingo samples appear to have substantial gene flow from dogs. It is not clear if these samples are included in the subsequent analysis (they are excluded from the NJ tree). Also, it is not clear if the apparent dingo populations are treated as one or modeled separately, or what effect this would have on the various conclusions.

Reply and revision: We just removed the two dingoes in the demographic analysis. However, it was unclear if these 2 dingoes had the gene flow from dogs. Thus, we performed the D statistic as the form of D (Dhole, European breed; Pop1, Pop2), where Pop1 was tested using all dog groups in turn and Pop2 was each dingo. In summary, a gene flow with European breeds is indicated when pop1 is NGSDs but when population 1 is any other dog population, there is no significant gene flow between European breeds and dingoes. Thus, we re-do the demographic analysis involving all individuals.

We have revised it accordingly and updated our demographic analysis.

Diversity of dingos (supp fig 4). It is not clear what the parameter “theta pi” is. The sample collection is very skewed as for example quite distinct wolf groups are considered as one group even though they split long ago. This statistic and the meaning of this analysis should be explained.

Reply and revision: “theta pi” is genetic diversity (θ_π). Thank you very much for pointing the inaccuracy of sample collection. We have revised and discussed it accordingly.

The G-PHOCS analysis should also consider effects of gene flow on the estimates, as they split times are likely to be very sensitive to this. The 2.2×10^{-9} site/year mutation rate also does not seem appropriate, with studies calibrating using ancient DNA seem to suggest $\sim 1.3 \times 10^{-9}$ per year. The effect of the assumption of mutation rate should be discussed in detail since the dating of the dingo split is a key part of the study.

Reply and revision: We considered effects of gene flow in the G-PhoCS analysis. We set the gene flow between southern East Asia dogs and Indonesian dogs, and found an event of gene flow from southern East Asia dogs to Indonesian dogs. We used the new mutation rate to estimate the time, and put results in Supplement (Supplementary table S4, S6). It is worth to note that the results from mitochondrial genome are consistent with the results when setting the mutation rate of 2.2×10^{-9} site/year.

It is also not clear which samples were used in the G-PHOCS analysis. All of the dingos together? Excluding the two with large admixture? Which samples for the other parts? How were the variants called for GPHOCS analysis? Just using the regular GATK VCF after VQSR? Overall the description of this analysis is lacking

Reply and revision: We revised the Materials and Methods accordingly and added all necessary information in the Supplement as well.

The selection analysis is also somewhat superficial. Outlier approaches based on statistics are used, but there is not any way to set expectations. Since the authors infer the demographic history, these should be utilized to set null expectations for selection, particularly given the apparent strong bottleneck in dingos. Such simulations are critical for interpreting the results.

Reply and revision: Thank you very much for the comments. We performed the simulations analysis according to the demographic history as the null expectation for all statistical approaches of selections (Supplementary Figure S11-S13).

We have revised the manuscript accordingly.

I think the GO enrichment also has several shortcomings. First, it seems that there is no correction for multiple hypothesis testing. It is also not clear which method was used for GO enrichment testing. Furthermore, these methods usually do not account for spatial structure of genes along chromosomes, gene size etc, and are not directly appropriate for genome based enrichment analysis.

Reply and revision: We redid the GO enrichment analysis by g:Profiler1 and performed the Benjamini-Hochberg as multiple hypothesis correction. As a result, we identified 3 functional classes that were significantly overrepresented.

We revised the manuscript accordingly.

More importantly, I do not think that the core claim of “reversed selection” is well supported by the data as presented. Could not the described *Fst* pattern simply reflect pressure in modern dogs that occurred after the dog-dingo split, rather than the ‘reverse selection’ argument made by the

authors. Haplotype information should be very informative for such a scenario, in addition to the simple Fst in windows approach.

Reply and revision: To verify the 'reverse selection' argument, we performed PBS analysis instead of Fst according to reviewer 2's comment, and performed the haplotypes analysis as your suggestion based on the method described in Pendleton et al ². As a result, we found dingoes and gray wolf share the same pattern of haplotypes in these genes as shown in supplementary figure S14.

We have revised and discussed it accordingly.

Relatedly, many studies have attempted to find genome regions associated with dog domestication. For example, Pendleton et al recently used village dogs and ancient DNA to try to identify selection during domestication vs selection during later dog evolution. It would be valuable to compare the purported 'reversed selection' signals with the findings of that study.

Reply and revision: We have revised and discussed it accordingly.

References

1. Jüri R, Meelis K, Hedi P, Jaanus H, Jaak V. g:Profiler--a web-based toolset for functional profiling of gene lists from large-scale experiments. *Nucleic Acids Research* 2007, **35**(Web Server issue): W193-W200.
2. Pendleton AL, Shen F, Taravella AM, Emery S, Veeramah KR, Boyko AR, *et al.* Comparison of village dog and wolf genomes highlights the role of the neural crest in dog domestication. *Bmc Biology* 2018, **16**(1): 64.

Reviewer #2 (Remarks to the Author):

The authors present a study of the genomic mechanism of feralization, using the dingo, a canid species native to Australia, as a model in this manuscript. The manuscript presents novel insights into the feralization process, and presents good evidence for the role of certain behavior related genes in the process.

The article suggests that Indonesia, possibly through SE Asia, is the ancestral population of the dingos. Further, they suggest the presence of "reverse" selection on some of the genes they find to be outliers in Fst scans. These genes, according to the authors, underwent a second round of selection in the feralization process, to bestow upon the dingos traits that resemble those of wolves much more than the same traits in dogs. The methods and statistics used in these analyses need a closer look, as specific statistics (such as PBS and LSD) exist, exactly for the scenarios that the authors are examining here. I would highly recommend using these statistics instead of the adhoc Fst based comparisons used here. The functional verification of one of the putative genes under "reverse" selection is indeed a very nice part of the article, and kudos to the authors for following up on one of the most interesting genes of their set of findings. In terms of the scientific impact of the manuscript, the work presented here clarifies the geographic origin of the dingos. The authors

propose SE Asia as the source, with migration through the SE Asian islands, finally to colonize Australia. They dismiss the previous claims of an Austronesian human expansion as a possible source, based on some of the population history and demography dates. These dates are based on mutation rate estimates that are still not definitively agreed upon in the canid research community. Thus, in my opinion, the Austronesian expansion could still be a viable explanation.

Overall, the manuscript is well written and presented, but could definitely undergo a round of proof reading and copy editing, to improve some of the phrasing. There are some places where the sentences are a bit confusing.

Reply and revision: Thank you for your valuable comments. We have revised the manuscript accordingly and thoroughly. We have improved our language, and we performed the PBS instead of Fst. Besides, we have performed the simulation as null model to refine the results of selection. We have also performed outgroup f3 and treemix analysis for the relationship and gene flow among populations.

Specific comments:

Page 5: "After strict filtering, we identified ~24.7 million autosomal SNPs for further analysis." Insufficient details provided here. Please provide details on SNP calling and filtering. It is impossible to replicate these findings (or comment on their robustness without details). This is a recurrent theme, and the authors would do well to share more details on their analyses.

Reply and revision: We have added it in Materials and Methods.

Page 5: "In a two-dimensional plot of the genotypes, there is a clear separation in four groups: Tibetan wolves, other wolves, dogs and dingoes/NGSDs." In figure 1B, I see 3 of the four groups mentioned wolves, dogs and dingoes/NGSD. The lone Tibetan wolf clusters with the dogs - a bit of a surprise but can be explained by the low sample size (1 sample of Tibetan wolf), and the known oddities in the genetic history of the Tibetan wolf. I would suggest that the authors focus on the three groups where there is a clear signal of separation.

Reply and revision: We have revised it accordingly.

Page 5/6: "Among the dogs, Indonesian village dogs cluster closest to the dingoes/NGSDs, followed by indigenous dogs from southern East Asia (Fig. 1B). Notably, India and Taiwan have been suggested as possible origins for the dingo, but the dogs from these regions cluster relatively far from the dingoes." I suggest that the authors refrain from using the PCA and other dimension reduction representations to indicate origin of the dingo. They have much better and well supported evidence in both admixture and neighbor joining trees. First, the amount of variance explained by PC2 is not very high. Second, both the dog cluster and the dingo cluster are spread out on the PC2 axis, suggesting some level of genetic heterogeneity in these clusters, and therefore, the closeness of the Indonesian dogs might have resulted from gene flow in the (early) history of the dingoes, and especially the NGSD. I said earlier, better evidence is provided by some of their other analyses.

Reply and revision: We have added outgroup f3 and treemix analyses. The results of the outgroup

f3 test were put into Supplementary Information (Supplementary Table S2.). It suggested that the NGSDs are most closely related to dingoes, and then Indonesia dog, then indigenous dogs from southern East Asia. We have also performed the analysis of treemix, and put the results in Supplementary Information (Supplementary Figure S6, S7). It suggest that there is just one candidate admixture event, from the dingo/NGSD clade to the Papua New Guinea village dogs lineage.

We have rewritten the text accordingly.

Page 6:"The results indicate gene flow between the two dingoes (D05 and D06) and European breeds (D=0.1027 and 0.0827, Z =14.551 and 10.221) (Supplementary table 2)." The authors present convincing evidence that there is gene flow between D05, D06 and European breeds, but I would like to see the results of the same D statistic test performed on the other dingos as well (in supplementary table 2), if for nothing else, then just for comparative purposes.

Reply and revision: We have performed the analyses of gene flow on all dingoes with all possible combinations by D statistic in the form D (Dhole, European breed; Pop1, Pop2), where Pop1 was tested using all dog groups in turn and Pop2 was every dingo. In summary, a gene flow with European breads is indicated when Pop1 is NGSDs. However, when Pop1 is other dog populations, there is no significant gene flow indicated between European breads and dingoes. We have revised and discussed it accordingly (Supplementary Figure S3).

Page 7: The parameter " $\theta\pi$ " should have the " π " in subscript. "This suggests a severe bottleneck event in the evolutionary history of dingoes, which may explain the long phylogenetic branch."

Although a bottleneck is one possible explanation, there are other explanations that the authors should consider. Long periods of isolation with low effective population size, as was probably the case in the dingos, will lead to long external branches and low diversity. This is very reasonable alternative explanation to the severe bottleneck hypothesis.

Reply and revision: We have revised it accordingly.

Page 9: As shown by the authors and previous studies, there is clear evidence of geographic structure in the dingos. But there is little to no evidence of two introductions of the dingo into Australia, and the authors provide none, and do not test this hypothesis after presenting it.

Reply and revision: We have removed the statement.

Page 10: First paragraph. Needs some rewriting to clarify, esp. the sentence starting with "A window ...".

Page 10: Use of Fst between the dingos and the Indonesian/South East Asian dogs is a valid way to identify genomic regions that differ between these two groups, but this does not allow identifying these regions are being positively selected in specifically the dingo lineage. These genes might have undergone changes in the other lineage as well. Using iHS to add support to these windows does alleviate these issues, but it would be good to provide the distribution of iHS

window scores in the supplementary material

Reply and revision: We have rewritten this. We added the distribution of iHS window information in the supplementary material.

Page 10: Section "Genes with reversible selection in domestication and feralization" This section uses 3 different pairwise F_{st} measures to figure out the branch on which the selection might have occurred. A much more principled approach, which should be used here, is the Population Branch Statistic (PBS) which is designed to do exactly what the authors are trying to achieve here. I would highly recommend switching to the PBS statistic instead of this adhoc measure using three different F_{st} statistic. This would allow them to obtain an empirical distribution for the PBS statistic which then can be used to select outlier regions of the genome.

Page 11: "This suggests that these 14 genes evolved during both the feralization and the domestication steps, such that feralization imposed reversed selection compared to the domestication." The authors argue that the signature of high F_{st} between wolves and dogs, and low F_{st} between dingos and wolves shows that these genes underwent a selection in the reverse direction (to revert back to the wolf form) in the dingos. This is not the most parsimonious explanation for this observation. One possible explanation would be the selection on these genes in the South East Asian dogs after the split of the dingos. The authors propose that this split happened ~5000 years ago, giving plenty of time for this selection to have happened in the SE Asian dogs. The use of the PBS statistic would address these problems.

Reply and revision: Thank you very much for the above 2 comments. We calculated PBS2 accordingly. Based on the PBS1, we identify 87 candidate windows of feralization. Based on the second PBS, we found 17 candidate windows of feralization.

We revised the manuscript accordingly.

Page 13: "Interestingly, the demographic analysis indicates that the dingo ancestors originated from southern China 5,300-6,600 years ago, at which time large-scale rice farming was established in this region." The authors do provide evidence that Indonesian dogs are the ancestors of dingos, but this statement here suggests that Southern China was the ancestral population of the dingos, which is only marginally (at least without further analyses) supported in this manuscript. The lack of the *AMY2B* copy number increase in the dingos also suggests that these split from the SE Asian dogs before the copy number expansion in these dogs. If I am not mistaken, this is the point that the authors are trying to make in the last sentence of the paragraph. If so, I would suggest a rewrite to explicitly state this, since the connection to rice farming is currently very tenuous.

Reply and revision: We have revised and discussed it accordingly.

Supplementary figure 2: Please provide bootstrap/local posterior probability support values for the nodes in this tree.

Reply and revision: We have added it in supplementary.

Reviewers' Comments:

Reviewer #1:

Remarks to the Author:

The revision is improved, but I still think that aspects of the analysis are incomplete and overstated. In particular, the claim of "reversible selection" (which is given in the title) I feel remains speculative and the gene enrichment analyses are not reliable due to biases in gene size and colocation that do not appear to be accounted for and the plots of the highlighted examples raise additional questions. The demographic analyses is more convincing, but there are several points related to interpretation and methods that should be addressed. The claim of reversible selection on specific genes is prominently made but I think the current evidence is sufficiently strong on this key point.

I believe there is an alternative interpretation of the D statistics. The authors' perform analysis of D(Dhole, Europe, Pop1, Pop2) and find positive D values when Pop1 is NGSD and Pop2 is a Dingo, which is interpreted as evidence of gene flow from Europe breeds to Dingo. However, this positive D is only found when Pop1 is NGSD, not when village dogs from India or Asia are used. I think an alternative interpretation of the positive D value is gene flow from Dhole to NGSD. This seems more likely since other combinations of populations in Asia are not showing the positive value which would be expected if there was Europe -> Dingo gene flow. Further consideration of these possibilities is warranted.

Details of the G-PhoCS analysis still seem inadequate. Was the same variant call set (from combined data discovery) used? Single sample calling is often used in these analyses, rather than taking genotypes from the joint call set, but the details should be provided. Also, there are no credible intervals on any of the estimates, making comparisons challenging. It is typical to include an out group in G-PhoCS and I wonder why that was not done here. Use of comparable out groups would aid comparisons with the several other analyses of various aspects of dog history that have been published using these methods, and would lead to greater confidence in the reported results.

In the supplementary figures (S11, S12, S13) it is unusual to only show the top 1-5% of the real observed data. It would be more informative to show that whole data distribution in order to assess how well the observed data is fit with the simulated data. This can be a useful sign for the quality of the simulation in terms of capturing key factors.

Figure S14 seems key to supporting the claims for selection, but I have some challenges in understanding it. In the three given examples, the dingos seem to largely have haplotypes which match the wolves (mostly 'red'). Is this indicating selection of wolf haplotypes in these samples, as opposed to newly favored alleles that would have formed on the background of the dog haplotypes? What does this say in terms of the nature of the supposed 'feralization' in terms of regions targeted and their potential origin from wolf haplotypes? This gets to the major point of the authors claims and I still struggle to see how strong the evidence is in support of it.

Also, in many cases there seem to be dogs that are heterozygous ('blue') for large regions, which I would not expect, but perhaps my expectations are not correct given the samples used. Individual sample labels may help clarify.

A major focus is the functional enrichment analysis performed. Such efforts are notoriously challenging. The authors used gProfiler, which is based on an input list of genes. I believe that this does not account for factors such as gene size and clustering of related gene families – essentially, it assumes each gene is equally likely to be chosen. This assumption may be warranted in gene expression analyses, but is not robust in genome based approaches. Relying on such methods is a

limitation. A random choice of similarly distributed intervals would likely also lead to pathways that appear to be statistically enriched.

Reviewer #2:

Remarks to the Author:

The authors have addressed many of my concerns, and implemented new PBS analyses, which have added value to the paper. I still have concerns about the authors conclusions about reversible selection during feralization process in the dingos. I am not convinced by their use of PBS or iHS that there are genes undergoing reverse selection to the wolf ancestral state. To me, the most parsimonious explanation is still of the selection occurring on the other dog branch after split from the Dingos - a point also noted by reviewer 1. Further, their timing of the split, if we use the mutation rate of 1.3×10^{-9} per year /bp, is about 8800 years, which is before the introduction of rice farming in china, which would then account for the lack of AMY2B copy number expansion in the dingos. Further, their choice of mutation rates of 2.2×10^{-9} is a bit odd, since most recent papers prefer 1.3×10^{-9} . This recalibrates some of these dates and makes them in line with their mitochondrial findings. Their selection analyses and followup are nice, but do not prove the point that this is reversible selection. One suggestion would be to look at NGSD, and see if they share these mutations, since, as far as my rudimentary knowledge of the NGSD goes, they do not form packs and share the same social structures as dingos. With these caveats in mind, I do not think the article is completely ready for publishing.

Pg 2 Line 49-51: "Interestingly, we found that 13 genes have shifted allele frequencies compared to dogs but not compared to wolves. This suggests that the selection affecting these genes during domestication of the wolf was reversed in the feralization process." Again, the first statement of this argument, I have no trouble with, but the second part of it claiming that the selection was reversed is not supported by anything that the authors present, so I would not suggest calling it reversed selection.

Pg 3, Line 84-87: "nterestingly, we found that 13 genes have shifted allele frequencies compared to dogs but not compared to wolves. This suggests that the selection affecting these genes during domestication of the wolf was reversed in the feralization process." *****

Pg 7, Line 170-183: The interpretation of the D statistics provided by the authors does not correctly represent the results. Their results, where the D-stats were significantly positive for some of the dingos, when they used the NGSD as pop1 indicate the that the Dingos have a significantly higher proportion of European breeds than NGSD. Further, their findings of -ve Z scores when using other dogs as pop1 means that all the other tested dogs are genetically closer to European breeds than the dingos. Maybe including some European breeds in to the PCA and admixture would clear this up. This does cast some doubt on their statement that there is no significant gene flow from European breeds into Dingos.

Pg 10 Line 279-281: The split time estimate from mtDNA of ~4500 years split from NGSD is not the same as the nuclear genome estimate of split from Indonesian dogs. This is almost more consistent with the 1.3×10^{-9} mutation rate estimates.

Pg 11 Line 295-297: PBS can be calibrated using the empirical distribution to choose the top x% (here they use 5%) of the PBS findings to follow up on as being interesting for selection in the specific branch. PBS distribution (genome wide) take the Thus, I am not sure why the authors are calculating

the PBS statistic on simulated data. Further, if they are going to use simulations here, they definitely need to provide more details, since one has to now also simulate the population history of wolves and other dogs as well.

Pg 12 Line 332-338: Here again, the authors conclusions are a bit difficult to follow. The genes/regions that they focus on - genes with high PBS1 and iHS values and high PBS2 value. First, high PBS1 and iHS values indicate high drift on the Dingo branch, implying selection in Dingos. Further, high PBS2 indicates long drift length in South East Asian / South China dogs, so selected in these dogs. So regions/genes which fulfill all three criteria are going to be ones which have high selection in Dingos (but not in SE Asian/S china dogs, compared to European breeds), and also high selection in SC dogs (but not dingos compared to wolves as outgroups). I am not sure how combining these implies a gene that went through reverse selection to being similar to wolves. The most parsimonious explanation is still the ones suggested by reviewer 1, with that being, selection in non-dingo dogs after their split from Dingos. I still do not see sufficient evidence that there was "reverse" selection in Dingos, reverting them back to the wolf form. A further attempt would be to look at the state of the same "reverse selected" genes in the NGSD, to see if there was a selection event specific to Dingos, or if the NGSD also share the same states at these genes.

Pg 13 Line 400-405: There are other dogs that do not show the expanded AMY2B copy numbers, such as the sled dogs, and the northern dog lineages such as the husky, and the malamute. So the diet change might have not had an impact on the dingos genome, but it might be that the copy number expansion did not happen yet. Also, if we use the 1.3×10^{-9} mutational rate, then the split of the dingos from the chinese dogs would predate rice farming in China.

Reviewer #1 (Remarks to the Author):

The revision is improved, but I still think that aspects of the analysis are incomplete and overstated. In particular, the claim of “reversible selection” (which is given in the title) I feel remains speculative and the gene enrichment analyses are not reliable due to biases in gene size and colocation that do not appear to be accounted for and the plots of the highlighted examples raise additional questions. The demographic analyses is more convincing, but there are several points related to interpretation and methods that should be addressed. The claim of reversible selection on specific genes is prominently made but I think the current evidence is sufficiently strong on this key point.

Reply and revision: Thank you very much for your advice. We have removed the claim of the reversible selection. Besides, we have added more details in demographic analyses, and have changed the method of enrichment analysis to topGO to control for biasing factors such as gene size and clustering of related gene families.

We have revised the manuscript accordingly and hope it will meet your criterion.

I believe there is an alternative interpretation of the D statistics. The authors’ perform analysis of D(Dhole, Europe, Pop1, Pop2) and find positive D values when Pop1 is NGSD and Pop2 is a Dingo, which is interpreted as evidence of gene flow from Europe breeds to Dingo. However, this positive D is only found when Pop1 is NGSD, not when village dogs from India or Asia are used. I think an alternative interpretation of the positive D value is gene flow from Dhole to NGSD. This seems more likely since other combinations of populations in Asia are not showing the positive value which would be expected if there was Europe -> Dingo gene flow. Further consideration of these possibilities is warranted.

Reply: You are right that there could be the gene flow from Dhole to NGSD. Thus, we used the Fox as the outgroup and got the same result that there are no gene flow. Besides, we added PCA and admixture analysis just using European breeds and Dingoes according to the reviewer 2’s comment. The PCA plots shows that D05 and D06 are close to European breeds in this PCA, and the admixture revealed that the two dingoes have the signal of mixture with European breed dogs. We put all these analysis in the Results.

Details of the G-PhoCS analysis still seem inadequate. Was the same variant call set (from combined data discovery) used? Single sample calling is often used in these analyses, rather than taking genotypes from the joint call set, but the details should be provided. Also, there are no credible intervals on any of the estimates, making comparisons challenging. It is typical to include an out group in G-PhoCS and I wonder why that was not done here. Use of comparable out groups would aid comparisons with the several other analyses of various aspects of dog history that have been published using these methods, and would lead to greater confidence in the reported results.

Reply: We used the same variant call set, and we have added those details and credible intervals in the Results. In the current study, we focused on the demographic history of dingoes and NGSDs, thus, the dogs from South East Asia can be the out groups as the closest clades. There are many published studies with the same strategies^{1, 2, 3, 4}.

In the supplementary figures (S11, S12, S13) it is unusual to only show the top 1-5% of the real

observed data. It would be more informative to show that whole data distribution in order to assess how well the observed data is fit with the simulated data. This can be a useful sign for the quality of the simulation in terms of capturing key factors.

Reply: We have revised it accordingly.

Figure S14 seems key to supporting the claims for selection, but I have some challenges in understanding it. In the three given examples, the dingos seem to largely have haplotypes which match the wolves (mostly 'red'). Is this indicating selection of wolf haplotypes in these samples, as opposed to newly favored alleles that would have formed on the background of the dog haplotypes? What does this say in terms of the nature of the supposed 'feralization' in terms of regions targeted and their potential origin from wolf haplotypes? This gets to the major point of the authors claims and I still struggle to see how strong the evidence is in support of it.

Reply: We agree with you that there is currently insufficient evidence without the ancient state of dingoes. Therefore, we removed these expressions and revised the whole manuscript accordingly.

Also, in many cases there seem to be dogs that are heterozygous ('blue') for large regions, which I would not expect, but perhaps my expectations are not correct given the samples used. Individual sample labels may help clarify.

Reply: We have revised it accordingly.

A major focus is the functional enrichment analysis performed. Such efforts are notoriously challenging. The authors used gProfiler, which is based on an input list of genes. I believe that this does not account for factors such as gene size and clustering of related gene families – essentially, it assumes each gene is equally likely to be chosen. This assumption may be warranted in gene expression analyses, but is not robust in genome based approaches. Relying on such methods is a limitation. A random choice of similarly distributed intervals would likely also lead to pathways that appear to be statistically enriched.

Reply: We have revised the method of enrichment analysis to the topGO, and calculated permutation-based p values to control the biasing factors such as gene size and clustering of related gene families. As results, we identified 67 GO terms that were significantly overrepresented (pperm <0.05).

We have revised the manuscript accordingly.

References

1. Frantz LA, Mullin VE, Pionnier-Capitan M, Lebrasseur O, Ollivier M, Perri A, *et al.* Genomic and archaeological evidence suggest a dual origin of domestic dogs. *Science* 2016, **352**(6290): 1228.
2. Skoglund P, Ersmark E, Palkopoulou E, Dalén L. Ancient Wolf Genome Reveals an Early Divergence of Domestic Dog Ancestors and Admixture into High-Latitude Breeds. *Current Biology* 2015, **25**(11): 1515-1519.
3. Sacks BN, Brown SK, Stephens D, Pedersen NC, Wu JT, Berry O. Y chromosome analysis of dingoes and southeast asian village dogs suggests a neolithic continental expansion from

Southeast Asia followed by multiple Austronesian dispersals. *Mol Biol Evol* 2013, **30**(5): 1103-1118.

4. Cairns KM, Wilton AN. New insights on the history of canids in Oceania based on mitochondrial and nuclear data. *Genetica* 2016, **144**(5): 553-565.

Reviewer #2 (Remarks to the Author):

The authors have addressed many of my concerns, and implemented new PBS analyses, which have added value to the paper. I still have concerns about the authors conclusions about reversible selection during feralization process in the dingos. I am not convinced by their use of PBS or iHS that there are genes undergoing reverse selection to the wolf ancestral state. To me, the most parsimonious explanation is still of the selection occurring on the other dog branch after split from the Dingos - a point also noted by reviewer 1. Further, their timing of the split, if we use the mutation rate of 1.3×10^{-9} per year /bp, is about 8800 years, which is before the introduction of rice farming in china, which would then account for the lack of AMY2B copy number expansion in the dingos. Further, their choice of mutation rates of 2.2×10^{-9} is a bit odd, since most recent papers prefer 1.3×10^{-9} . This recalibrates some of these dates and makes them in line with their mitochondrial findings.

Their selection analyses and followup are nice, but do not prove the point that this is reversible selection. One suggestion would be to look at NGSD, and see if they share these mutations, since, as far as my rudimentary knowledge of the NGSD goes, they do not form packs and share the same social structures as dingos. With these caveats in mind, I do not think the article is completely ready for publishing.

Reply and revision: Thank you very much for your advice. We have removed the point of the reversible selection. We have replaced the mutation rate to 1.3×10^{-9} per year and have added more details in the admixture and demographic analyses.

We have revised the manuscript accordingly and thoroughly and hope our revision could meet your criterion.

Pg 2 Line 49-51: "Interestingly, we found that 13 genes have shifted allele frequencies compared to dogs but not compared to wolves. This suggests that the selection affecting these genes during domestication of the wolf was reversed in the feralization process." Again, the first statement of this argument, I have no trouble with, but the second part of it claiming that the selection was reversed is not supported by anything that the authors present, so I would not suggest calling it reversed selection.

Line 84-87: "nterestingly, we found that 13 genes have shifted allele frequencies compared to dogs but not compared to wolves. This suggests that the selection affecting these genes during domestication of the wolf was reversed in the feralization process." *****

Reply and revision: We removed the related contents.

Pg 7, Line 170-183: The interpretation of the D statistics provided by the authors does not correctly represent the results. Their results, where the D-stats were significantly positive for some of the dingos, when they used the NGSD as pop1 indicate that the Dingos have a significantly higher proportion of European breeds than NGSD. Further, their findings of -ve Z scores when using other dogs as pop1 means that all the other tested dogs are genetically closer to European breeds than the dingos. Maybe including some European breeds in to the PCA and admixture would clear this up. This does cast some doubt on their statement that there is no significant gene flow from European breeds into Dingos.

Reply and revision: We have added the PCA and admixture analysis just using European breeds and Dingoes accordingly. Both results show dingoes and European breeds have a clear separation, but D05 and D06 are close to European breeds in the PCA plot, and the two dingoes also show a mixture with European breed dogs in admixture analysis.

We have added the analysis in the results.

10 Line 279-281: The split time estimate from mtDNA of ~4500 years split from NGSD is not the same as the nuclear genome estimate of split from Indonesian dogs. This is almost more consistent with the 1.3×10^{-9} mutation rate estimates.

Reply and revision: We have replaced the mutation rate.

g 11 Line 295-297: PBS can be calibrated using the empirical distribution to choose the top x% (here they use 5%) of the PBS findings to follow up on as being interesting for selection in the specific branch. PBS distribution (genome wide) take the Thus, I am not sure why the authors are calculating the PBS statistic on simulated data. Further, if they are going to use simulations here, they definitely need to provide more details, since one has to now also simulate the population history of wolves and other dogs as well.

Reply and revision: We performed the simulations analysis according to the demographic history as the null expectation for all statistical approaches of selection. Since our demographic analysis did not include wolves and European breeds, the effective population sizes and split time for European breeds were taken from Liu et al¹ and Wang et al².

We have added the detail in the Materials and Methods.

Pg 12 Line 332-338: Here again, the authors conclusions are a bit difficult to follow. The genes/regions that they focus on - genes with high PBS1 and iHS values and high PBS2 value. First, high PBS1 and iHS values indicate high drift on the Dingo branch, implying selection in Dingos. Further, high PBS2 indicates long drift length in South East Asian / South China dogs, so selected in these dogs. So regions/genes which fulfill all three criteria are going to be ones which have high selection in Dingos (but not in SE Asian/S china dogs, compared to European breeds), and also high selection in SC dogs (but not dingos compared to wolves as outgroups). I am not sure how combining these implies a gene that went through reverse selection to being similar to wolves. The most parsimonious explanation is still the ones suggested by reviewer 1, with that being, selection in non-dingo dogs after their split from Dingos. I still do not see sufficient

evidence that there was "reverse" selection in Dingos, reverting them back to the wolf form. A further attempt would be to look at the state of the same "reverse selected" genes in the NGSD, to see if there was a selection event specific to Dingos, or if the NGSD also share the same states at these genes.

Reply and revision: we did not use PBS2 to get the signal of selection, but to evaluate the similarity between dog, wolf, and dingo. The high value of PBS2 represents the region that dingoes are more like grey wolves compared to domestic dogs. And we have removed the claim of reversible selection. We also looked into the haplotypes of the selected genes in the NGSD, and found the haplotypes in NGSDs is same as those in dingoes in similar with these in grey wolves. We have revised the manuscript accordingly.

Pg 13 Line 400-405: There are other dogs that do not show the expanded AMY2B copy numbers, such as the sled dogs, and the northern dog lineages such as the husky, and the malamute. So the diet change might have not had an impact on the dingos genome, but it might be that the copy number expansion did not happen yet. Also, if we use the 1.3×10^{-9} mutatio rate, then the split of the dingos from the chinese dogs would predate rice farming in China.

Reply and revision: We have revised it accordingly and have replaced the mutation rate.

References

1. Liu YH, Wang L, Xu T, Guo X, Li Y, Yin TT, *et al.* Whole-genome sequencing of African dogs provides insights into adaptations against tropical parasites. *Molecular Biology & Evolution* 2017.
2. Wang GD, Zhai W, Yang HC, Wang L, Zhong L, Liu YH, *et al.* Out of southern East Asia: the natural history of domestic dogs across the world. *Cell research* 2016, **26**(1): 21-33.

Reviewers' Comments:

Reviewer #1:

Remarks to the Author:

I am still somewhat confused about the interpretation of the D stats. D05 and D06 are not drastically higher than the rest of the 7 dingo samples that are significantly higher. As reviewer 2 mentioned, is this not then evidence for gene flow from European dogs into most Dingos? Should this be accounted for? This goes to a major motivating claim for the absence of gene flow making Dingo a remarkable system for genetic analysis during feralisation.

Line 371 I am still also somewhat confused by the reasoning. For the sentence "selection during the later development of the European breed and that selection had not occurred for the ancestors of the dingos" – would it be right to add then that the claim is that there was selection in European breeds (but not Asian) at these regions, and that also there was selection in the dingos at these same region? It might clarify to be explicit in the reasoning. It has not clear how this connects to the point in lines 472-474, that selection only along dog lineage after the split from dingo is possible.

The title of the supplement seems to differ from revised title of the main paper

Reviewer #2:

Remarks to the Author:

The authors have made many of the changes suggested by both reviewers and have implemented many additional tests and included functional data. As the manuscript stands now, I am satisfied that it will contribute to new understanding of the history of dingos, thus filling in a piece that is currently missing from the evolutionary history of canids. Further, it will also undoubtedly be the basis of many future studies of feralization, both in canids and other species. There are still many places where the manuscript requires a final readthrough and some copyediting, but on the scientific front, I have only minor quibbles which I mention below.

Specific comments:

Lines 173-182: "Interestingly, the results ... European breeds and dingoes ($Z < -3$) (Supplementary Figure S3)."

Again, with respect to the interpretation of the d-stats, since only the NGSD show any signal relative to the dingos, when used in the D-stat set up, and replacing the NGSD with the other dogs results in no signal, one possible interpretation of the results is to say that the NGSD probably have some component outside of the European breed dogs, which is not found in the dingos. The interpretation that the dingos could have had gene flow from the European breeds does not make sense in light of the later parts of the same paragraph. Further, figure S3 does indeed show that there is no gene flow between Dingos and European breeds, when any dog other than the NGSD is used. But the finding is not that there is no significant gene flow at all. It actually is that all other dogs show higher affinity to European breed dogs than the dingos. A critical look at this paragraph, combined with a rewrite to make it clear is probably required. Further, repeating the whole D-stat framework with red fox instead of the dhole as the outgroup does not add any value, unless there was some doubt dholes being a valid outgroup.

Minor comment - European breads should read European breeds.

Lines 298-300: "As a result, the top 5% of the PBS1 based on simulated data ($PBS1 = 0.13736$) was lower than based on the real data ($PBS1 = 0.14476$) (Supplementary Figure S14)."

The authors, both in the ms and in reply to reviewer 2, where they say "We performed the simulations analysis according to the demographic history as the null expectation for all statistical approaches of selection.", defend the use of simulation using the estimated demography to figure out top 5% cutoffs for PBS and iHS statistics. (I understand that they are using the more stringent of the two - simulation and empirical values to figure out the cutoffs). I am opposed to the idea of using the simulations to play any role in the estimation of the cutoffs in this case, as a matter of principle. My objections stem from the fact that the using estimated demographies to find the cutoffs using simulations makes the inherent assumption that the demographic model fitted encompasses all the important demographic process that might affect the null distribution of the test statistics - in this case the PBS and iHS statistics. To rephrase, the presence of unmodelled demographic events will affect the null distribution of these statistics, and thus make the cutoffs obtained from the not well calibrated. This point is reinforced from their plots S14 and S16, and to a certain extent S15, where the observed distribution is very different from the simulated distribution throughout the range of the statistics. Thus I would stick to the use of the empirical values of the statistic to figure out the interesting genes. Another point to make is that these statistics are not being tested for significance in any sense, so the use of a null distribution is irrelevant in any case.

Lines 340-342: Usage of regions with high PBS2 to choose regions where the dingo and wolf are similar to each other but the other dogs are different from these two. The authors, both in the manuscript and the response to reviewer 2, claim that they use high values of PBS2 to choose regions of the genome where the dingos and wolves are similar to each other, to the exclusion of other dogs. In the response to reviewer 2, they say "we did not use PBS2 to get the signal of selection, but to evaluate the similarity between dog, wolf, and dingo. The high value of PBS2 represents the region that dingoes are more like grey wolves compared to domestic dogs.". But this is a bit problematic. The PBS2 statistic that is described will find regions of the genome where other dogs have long branch lengths from the root of the triple - dogs, dingos, wolves. Thus, there might be many regions of the genome where dingos and wolves are closer to each other, but there is no selection on the other dogs branch, thus leading to a small PBS2 signal. In this particular case, the use of the PBS2 signal only leads to loss of power for the authors case and not to false positives, so not much harm has been done.

Lines 558-560: "The mutation rate ... genome estimate."

The period after "studies" should be removed, and the last clause should read "and the result when we use this as the mutation rate is in agreement with the mitochondrial genome estimate."

Table S2: A block jack-knife style estimate of the f_3 statistic would be very helpful in interpreting the differences in the values of the statistic.

Reviewer #1 (Remarks to the Author):

I am still somewhat confused about the interpretation of the D stats. D05 and D06 are not drastically higher than the rest of the 7 dingo samples that are significantly higher. As reviewer 2 mentioned, is this not then evidence for gene flow from European dogs into most Dingos? Should this be accounted for? This goes to a major motivating claim for the absence of gene flow making Dingo a remarkable system for genetic analysis during feralisation.

Reply and revision: Thank you very much for your advice. We have added a more rigorous analysis of D statistics in the form of D (Dhole, European breed; Pop1, Pop2), where Pop1 and Pop2 was all individual dingo tested in turn (method similar to Cairns et al. and Wang et al.^{1,2}), and the results show that when Pop2 was three of the dingoes (D01, D05 and D06) there was significantly positive D ($Z > 3$) in most combinations and no significantly negative D ($Z < -3$) in any comparison. This suggests that the three dingoes may have had gene flow with European breeds. We have clarified in the text that, after the isolation during the feralization, hybridization between dingo and dog occurs since the arrival of Europeans 200 years ago. We have revised the manuscript accordingly and hope it will meet your criterion.

Line 371 I am still also somewhat confused by the reasoning. For the sentence “selection during the later development of the European breed and that selection had not occurred for the ancestors of the dingos” – would it be right to add then that the claim is that there was selection in European breeds (but not Asian) at this regions, and that also there was selection in the dingos at these same region? It might clarify to be explicit in the reasoning. It has not clear how this connects to the point in lines 472-474, that selection only along dog lineage after the split from dingo is possible.

Reply and revision: We agree with your view, and we have complemented the text accordingly.

The title of the supplement seems to differ from revised title of the main paper

Reply and revision: We have revised it accordingly.

Reviewer #2 (Remarks to the Author):

The authors have made many of the changes suggested by both reviewers and have implemented many additional tests and included functional data. As the manuscript stands now, I am satisfied that it will contribute to new understanding of the history of dingos, thus filling in a piece that is currently missing from the evolutionary history of canids. Further, it will also undoubtedly be the basis of many future studies of feralization, both in canids and other species. There are still many places where the manuscript requires a final readthrough and some copyediting, but on the scientific front, I have only minor quibbles which I mention below.

Reply and revision: Thank you very much for your advice. We have made a new D statistics, and the new result show that three dingoes may have gene flow with breeds. We no longer used the simulation analysis to find the threshold in the main text but keep it in Supplemental Information. We have revised the manuscript with your advice and hope it will meet your criterion.

Specific comments:

Lines 173-182: "Interestingly, the results ... European breeds and dingoes ($Z < -3$) (Supplementary Figure S3)."

Again, with respect to the interpretation of the d-stats, since only the NGSD show any signal relative to the dingoes, when used in the D-stat set up, and replacing the NGSD with the other dogs results in no signal, one possible interpretation of the results is to say that the NGSD probably have some component outside of the European breed dogs, which is not found in the dingoes. The interpretation that the dingoes could have had gene flow from the European breeds does not make sense in light of the later parts of the same paragraph. Further, figure S3 does indeed show that there is no gene flow between Dingoes and European breeds, when any dog other than the NGSD is used. But the finding is not that there is no significant gene flow at all. It actually is that all other dogs show higher affinity to European breed dogs than the dingoes. A critical look at this paragraph, combined with a rewrite to make it clear is probably required. Further, repeating the whole D-stat framework with red fox instead of the dhole as the outgroup does not add any value, unless there was some doubt dholes being a valid outgroup.

Minor comment - European breeds should read European breeds.

Reply and revision: Thank you very much for your advice. We have added a more rigorous analysis of D statistics in the form of D (Dhole, European breed; Pop1, Pop2), where Pop1 and Pop2 was all dingo individuals tested in turn (method similar to Cairns et al. and Wang et al^{1,2}). The results show that when Pop2 are three of the dingoes (D01, D05 and D06) they had significantly positive D ($Z > 3$) in most combinations, suggesting that the three dingoes could have a gene flow with European breeds. We revised the manuscript accordingly.

Lines 298-300: "As a result, the top 5% of the PBS1 based on simulated data ($PBS1 = 0.13736$) was lower than based on the real data ($PBS1 = 0.14476$) (Supplementary Figure S14)."

The authors, both in the ms and in reply to reviewer 2, where they say "We performed the simulations analysis according to the demographic history as the null expectation for all statistical

approaches of selection.", defend the use of simulation using the estimated demography to figure out top 5% cutoffs for PBS and iHS statistics. (I understand that they are using the more stringent of the two - simulation and empirical values to figure out the cutoffs). I am opposed to the idea of using the simulations to play any role in the estimation of the cutoffs in this case, as a matter of principle. My objections stem from the fact that the using estimated demographies to find the cutoffs using simulations makes the inherent assumption that the demographic model fitted encompasses all the important demographic process that might affect the null distribution of the test statistics - in this case the PBS and iHS statistics. To rephrase, the presence of unmodelled demographic events will affect the null distribution of these statistics, and thus make the cutoffs obtained from the not well calibrated. This point is reinforced from their plots S14 and S16, and to a certain extent S15, where the observed distribution is very different from the simulated distribution throughout the range of the statistics. Thus I would stick to the use of the empirical values of the statistic to figure out the interesting genes. Another point to make is that these statistics are not being tested for significance in any sense, so the use of a null distribution is irrelevant in any case.

Reply and revision: We have revised it accordingly. We have used the empirical threshold and the results are consistent.

Lines 340-342: Usage of regions with high PBS2 to choose regions where the dingo and wolf are similar to each other but the other dogs are different from these two. The authors, both in the manuscript and the response to reviewer 2, claim that they use high values of PBS2 to choose regions of the genome where the dingos and wolves are similar to each other, to the exclusion of other dogs. In the response to reviewer 2, they say "we did not use PBS2 to get the signal of selection, but to evaluate the similarity between dog, wolf, and dingo. The high value of PBS2 represents the region that dingoes are more like grey wolves compared to domestic dogs.". But this is a bit problematic. The PBS2 statistic that is described will find regions of the genome where other dogs have long branch lengths from the root of the triple - dogs, dingos, wolves. Thus, there might be many regions of the genome where dingos and wolves are closer to each other, but there is no selection on the other dogs branch, thus leading to a small PBS2 signal. In this particular case, the use of the PBS2 signal only leads to loss of power for the authors case and not to false positives, so not much harm has been done.

Reply and revision: We agree with your view, and we have revised the text accordingly.

Lines 558-560: "The mutation rate ... genome estimate."

The period after "studies" should be removed, and the last clause should read "and the result when we use this as the mutation rate is in agreement with the mitochondrial genome estimate."

Reply and revision: We have revised it accordingly.

Table S2: A block jack-knife style estimate of the f_3 statistic would be very helpful in interpreting the differences in the values of the statistic.

Reply and revision: We have added it accordingly.

1. Cairns KM, Shannon LM, Koler-Matznick J, Ballard JWO, Boyko AR. Elucidating biogeographical patterns in Australian native canids using genome wide SNPs. *Plos One* 2018, **13**(6): e0198754-.
2. Wang X, Zhou B-W, Yang MA, Yin T-T, Chen F-L, Ommeh SC, *et al.* Canine transmissible venereal tumor genome reveals ancient introgression from coyotes to pre-contact dogs in North America. *Cell research* 2019, **29**(7): 592-595.

Reviewers' Comments:

Reviewer #1:

None

Reviewer #2:

Remarks to the Author:

I want to thank the authors for a very nice finished product. They have addressed all the concerns I had regarding the manuscript over the rounds of reviews. As I have mentioned before, this article will make significant contributions to dingo, canid and feralization research. Thus, I am happy to recommend the article for publication in its current state.